# A novel, ataxic mouse model of ataxia telangiectasia caused by a clinically relevant nonsense mutation

Harvey Perez[1†], May F Abdallah[1†], Jose I Chavira[1†], Angelina S Norris[1], Martin T Egeland[1], Karen L Vo[1], Callan L Buechsenschuetz[1], Valentina Sanghez[1], Jeannie L Kim[1], Molly Pind[2], Kotoka Nakamura[3], Geoffrey G Hicks[2], Richard A Gatti[3], Joaquin Madrenas[1,4], Michelina Iacovino[1,5], Peter J McKinnon[6], Paul J Mathews[1,7]*

[1]The Lundquist Institute for Biomedical Innovation, Harbor-UCLA Medical Center, Torrance, United States; [2]Department of Biochemistry and Medical Genetics, Max Rady College of Medicine, University of Manitoba, Manitoba, Canada; [3]Department of Pathology & Laboratory Medicine, David Geffen School of Medicine, Los Angeles, United States; [4]Department of Medicine, Harbor-UCLA Medical Center, Torrance, United States; [5]Department of Pediatrics, Harbor-UCLA Medical Center, Torrance, United States; [6]Center for Pediatric Neurological Disease Research, St. Jude Pediatric Translational Neuroscience Initiative, St. Jude Children's Research Hospital, Memphis, United States; [7]Department of Neurology, Harbor-UCLA Medical Center, Torrance, United States

*For correspondence: pmathews@ucla.edu

[†]These authors contributed equally to this work

Competing interest: The authors declare that no competing interests exist.

**Abstract** Ataxia Telangiectasia (A-T) and Ataxia with Ocular Apraxia Type 1 (AOA1) are devastating neurological disorders caused by null mutations in the genome stability genes, A-T mutated (*ATM*) and Apratxin (*APTX*), respectively. Our mechanistic understanding and therapeutic repertoire for treating these disorders are severely lacking, in large part due to the failure of prior animal models with similar null mutations to recapitulate the characteristic loss of motor coordination (i.e., ataxia) and associated cerebellar defects. By increasing genotoxic stress through the insertion of null mutations in both the *Atm* (nonsense) and *Aptx* (knockout) genes in the same animal, we have generated a novel mouse model that for the first time develops a progressively severe ataxic phenotype associated with atrophy of the cerebellar molecular layer. We find biophysical properties of cerebellar Purkinje neurons (PNs) are significantly perturbed (e.g., reduced membrane capacitance, lower action potential [AP] thresholds, etc.), while properties of synaptic inputs remain largely unchanged. These perturbations significantly alter PN neural activity, including a progressive reduction in spontaneous AP firing frequency that correlates with both cerebellar atrophy and ataxia over the animal's first year of life. Double mutant mice also exhibit a high predisposition to developing cancer (thymomas) and immune abnormalities (impaired early thymocyte development and T-cell maturation), symptoms characteristic of A-T. Finally, by inserting a clinically relevant nonsense-type null mutation in *Atm*, we demonstrate that **S**mall **M**olecule **R**ead-**T**hrough (SMRT) compounds can restore ATM production, indicating their potential as a future A-T therapeutic.

## Editor's evaluation

The authors have developed a new mouse model for Ataxia Telangiectasia. This mouse is of great value, which exhibits core features of the human disease including cerebellar Purkinje cells defects,

cancer, and immune abnormalities. The potential value of the model is demonstrated using small molecular weight compounds to restore molecular function.

## Introduction

Ataxia Telangiectasia (A-T) is a rare (one in ~100,000) (*Swift et al., 1986*), autosomal recessive genetic disorder characterized by cancer predisposition, immune deficiency, and a highly penetrant progressive and severe ataxia linked to cerebellar atrophy (*Rothblum-Oviatt et al., 2016*; *Boder and Sedgwick, 1958*; *Levy and Lang, 2018*). A-T patients typically die in their second or third decade of life (*Crawford et al., 2006*) from lymphatic cancers, respiratory infections, or complications of ataxia—unfortunately, survivability has not dramatically changed since the 1950s (*Micol et al., 2011*; *Rothblum-Oviatt et al., 2016*). While disease progression and cause of death vary widely across patients, the highly penetrant progressive decline in motor coordination is reported as having the greatest negative impact on a patient's quality of life (*Jackson et al., 2016*). Care is generally palliative, directed at reducing, limiting, or eliminating cancers or infections. No long-term therapies are available for treating the ataxia and associated cerebellar dysfunction and atrophy.

A-T is caused by deficiency or dysfunction of the ATM (A-T mutated) protein (*Savitsky et al., 1995*). Premature termination codon (PTC) causing nonsense mutations account for up to a half of known cases, with missense and deletion mutations also contributing (*Concannon and Gatti, 1997*; *Sandoval et al., 1999*). ATM is a serine/threonine PIKK family kinase that plays a key role in the DNA damage response (DDR), protecting cells from the tens of thousands of DNA lesions incurred each day (*Lindahl and Barnes, 2000*; *Kastan and Bartek, 2004*; *Shiloh and Ziv, 2013*). In the active monomeric form, ATM phosphorylates several key proteins halting the production of new DNA (cell cycle arrest) (*Ando et al., 2012*), and then, depending on severity of the damage, initiates DNA repair or programmed cell death (i.e., apoptosis) (*Ando et al., 2012*; *Rashi-Elkeles et al., 2006*). Several downstream DDR pathway targets of ATM have been identified, including p53, CHK2, BRCA1, SMC1, and NBS1 (*Matsuoka et al., 2007*). ATM's role in DNA repair is also implicated in normal immune system development, where it is proposed to contribute to the recombination of natural DNA splicing that occurs during gene rearrangement in T- and B-lymphocyte maturation (*Chao et al., 2000*; *Matei et al., 2006*; *Vacchio et al., 2007*; *Schubert et al., 2002*). Although its roles are still emerging, ATM has also been implicated in oxidative stress homeostasis (*Guo et al., 2010*) and mitophagy (*Valentin-Vega and Kastan, 2012*; *Pizzamiglio et al., 2020*).

It is unclear why ATM deficiency causes ataxia, but it is far from the only DDR protein linked to ataxia, as Aprataxin (APTX) (*Aicardi et al., 1988*), Meiotic recombination 11 homolog 1 (MRE11) (*Sedghi et al., 2018*), Nibrin (NBS1) (*van der Burgt et al., 1996*), Senataxin (SETX) (*Moreira et al., 2004*), and Tyrosyl-DNA Phosphodiesterase 1 (TDP1) (*Takashima et al., 2002*) when absent or dysfunctional can cause cerebellar-related ataxia. This suggests that the neurological features of genome instability syndromes have a common underlying cause, although this has yet to be mechanically demonstrated (*McKinnon, 2009*; *Rass et al., 2007*).

A major factor limiting our ability to define why the loss of DDR proteins, like ATM, selectively impacts the cerebellum and causes progressive ataxia is the lack of an animal model that recapitulates these neurological symptoms (*Lavin, 2013*). Several A-T rodent models have been created over the past several years by inserting gene mutations that cause protein dysfunction (lack kinase activity) or complete deficiency (*Herzog et al., 1998*; *Xu and Baltimore, 1996*; *Elson et al., 1996*; *Spring et al., 2001*; *Campbell et al., 2015*; *Quek et al., 2016*; *Tal et al., 2018*; *Lavin, 2013*); a minipig was also recently reported (*Beraldi et al., 2017*). However, none develop an overt, progressive ataxia with cerebellar dysfunction and atrophy that recapitulates the human disease, even though other aspects of the disorder like thyroid cancers, infertility, and immune abnormalities do develop. It remains unclear why these prior animal models fail to display the progressive ataxic phenotype (*Lavin, 2013*). It is possible that species-specific molecular compensations in mice provide redundancies or alternative pathways minimizing the effects of ATM deficiency in the brain (*El-Brolosy and Stainier, 2017*). It is also possible that the shortened lifespan of prior models (*Barlow et al., 1996*) is too brief for the stochastic mechanisms driving cerebellar dysfunction and atrophy to accumulate and impact motor behavior. Other challenges include potentially leaky genetic manipulations that result in low levels of ATM protein or active fragments with residual

kinase activity, thus limiting neuropathology (*Li et al., 2011*). The impact of missing such a crucial animal model has been significant, severely limiting experimental studies from identifying the cellular and molecular mechanisms and hampering pre-clinical development and testing of much needed therapeutics.

We test here whether increasing genotoxic stress, by placing null mutations in not just the *Atm* gene, but also the related *Aptx* gene, leads to a more representative mouse model that displays cerebellar dysfunction, atrophy, and the development of progressive ataxia. We chose to additionally knockout *Aptx* because its deficiency causes an A-T like disorder in humans called ataxia with ocular apraxia type 1 (AOA1), which does not feature A-T's other system defects that could increase the potential for prenatal lethality or early death (e.g., immunodeficiency and cancer predisposition) (*Coutinho et al., 2002*). Moreover, APTX is a phosphodiesterase involved in DNA reassembly after double- and single-stranded repair, and has a function downstream of—but not directly regulated by or related to—ATM (*Gueven et al., 2004*; *Schellenberg et al., 2015*; *Ahel et al., 2006*). We hypothesized that the functional expression of both proteins would have an additive effect and induce neurological dysfunction. Our results indeed demonstrate that mice deficient in ATM and APTX develop cerebellar dysfunction, atrophy, and a progressive and profound ataxia, while mice deficient in either protein alone do not. Additionally, double mutants displayed other characteristic symptoms of A-T, including defects in immune maturation and a high incidence of cancer (thymomas), making it the most representative model, from a phenotypic standpoint, to date.

Finally, we designed this new mouse model to test our recently developed **S**mall **M**olecule **R**ead-**T**hrough Compounds (SMRT) that enable translation through PTCs (*Du et al., 2013*). Thus, we inserted a PTC-causing nonsense mutation (103C>T) in the *Atm* gene common to a large family of North African A-T patients (*Gilad et al., 1996a*). This mutation results in a PTC at what would normally be amino acid 35 and the loss of ATM translation. Here, we report proof-of-principle experiments demonstrating that clinically relevant genetic mutations incorporated into the A-T mouse model are amenable to read-through compounds and thus appropriate for preclinical testing of SMRT compounds.

## Results
### Creation of a new A-T mutant mouse model expressing a clinically relevant nonsense mutation

To create a more clinically relevant mouse model of A-T, we used a gateway recombination cloning and site-directed mutagenesis method to recapitulate a c.103C>T (p.R35X) mutation in the *ATM* gene found in a large population of North African A-T patients (*Figure 1A* and Materials and methods) (*Gilad et al., 1996b*). The insertion of thymine in place of cytosine at this site in exon 3 results in a PTC-causing nonsense mutation in the ATM gene. Since the c.103C>T mutation results in different PTCs in the human gene compared to the mouse *Atm* gene—TGA versus TAG, respectively—we created two different mice by exchanging the mouse *Atm* exon 3 with either a human or mouse exon 3 variant with the c.103C>T mutation (*Figure 1B*). In the human variant, a 103C>T mutation of the mouse codon, where the arginine (R) encoding codon (CGA) becomes a TGA stop codon, results in a mouse we denote as $Atm^{R35X}$ (officially $Atm^{Tm1.1(103CAG)TGAMfgc}$). In the mouse variant, the c.103C>T mutation transforms a glutamine (Q)-encoding CAG codon into a TAG stop codon and is denoted $Atm^{Q35X}$ (officially $Atm^{Tm1.1(103C)TMfgc}$). The presence of the PTC results in a loss of ATM expression, either reduced by about half in the heterozygote expressing one normal mouse copy of the *Atm* gene ($Atm^{R35X/+}$ or $Atm^{Q35X/+}$), or completely in the homozygote ($Atm^{R35X/R35X}$ or $Atm^{Q35X/Q35X}$) (*Figure 1C*).

$Atm^{R35X/R35X}$; $Aptx^{-/-}$ (double mutant) mice were created by first crossing single mutant $Atm^{R35X/R35X}$ (congenic on the C57BL/ 6J background) and $Aptx^{-/-}$ (mixed C57BL/ 6J and 129 background) mice to generate heterozygote $Atm^{R35X/+}$; $Aptx^{+/-}$. F1-5 littermate $Atm^{R35X/+}$; $Aptx^{+/-}$ were then crossed within litters to create sufficient numbers of the desired experimental and control genotypes to determine how the loss of different amounts of ATM and APTX affects the animal's phenotype (*Figure 1D*). Like prior ATM-deficient A-T mouse models, ATM or APTX deficiency alone did not result in mice with ataxia (*Videos 1 and 2*). However, deficiency in both proteins ($Atm^{R35X/R35X}$; $Aptx^{-/-}$) results in the development of a severe and progressively ataxic phenotype (*Figure 1E*, *Videos 3 and 4*).

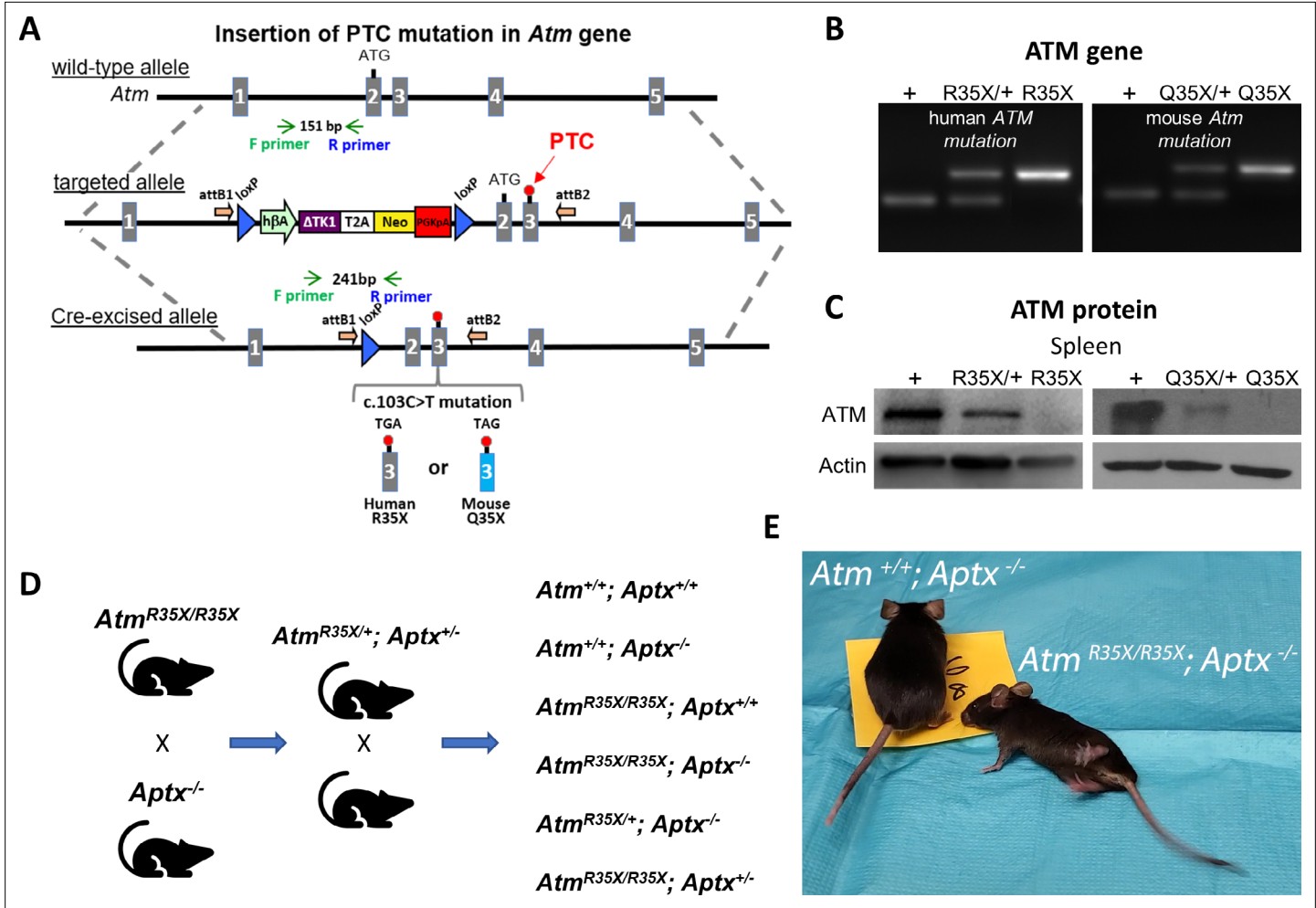

**Figure 1.** New A-T mouse models expressing clinically related PTCs. (**A**) The *Atm* gene locus was targeted by homologous recombination of a targeting vector containing a modified NorCOMM cassette in intron one and the corresponding A-T PTC mutation in exon 3 to create the targeted *Atm^R35X* and *Atm^Q35X* ES cell lines. Following germline transmission of these alleles in mice, the floxed NorCOMM cassette was removed by Cre-excision in vivo to produce the final *Atm^R35X* and *Atm^Q35X* mouse lines. (**B**) Genotyping of A-T mouse models. PCR agarose gel of mouse DNA shows 151 bp wild-type (+) allele band and 241 bp Cre-excised targeted allele band. (**C**) ATM levels were examined using immunoblot analyses of the spleen due to its high expression density in this tissue. Exemplar blots illustrate a gene dose effect of ATM protein expression in samples harvested from wild-type (+), heterozygous (R35X/+, Q35X/+), and homozygous *Atm^R35X/R35X* (R35X) and *Atm^Q35X/Q35X* (Q35X) mice as indicated. (**D**) Breeding scheme schematic for double mutant and control mice for this study. (**E**) *Atm^R35X/R35X; Aptx^−/−* mice develop an ataxia that at late stages results in a severe loss of motor coordination and ability to ambulate (see *Videos 1–4*). Abbreviations for panel 1: **hβA**-human beta Actin promotor; **ΔTK1**-delta TK1, inactivated Thymidine Kinase 1; (**T2A**)-self-cleaving peptide sequence; **Neo**-Neomycin gene; **PGKpA**-Phosphoglycerate kinase poly A tail; **loxP**-recombination elements are shown as a blue triangle; orientation of the Gateway **attB** recombination elements is shown by an orange arrow; orientation of the genotyping (F) and (R) primers is shown by green and blue arrows, respectively; and engineered PTC sites are shown in exon 3 by a red circle. A-T, Ataxia Telangiectasia; ATM, A-T mutated; PTC, premature termination codon.

The online version of this article includes the following source data for figure 1:

**Source data 1.** Original blots.

## ATM-deficient mice have lowered survivability and a high incidence of thymomas

We assessed the general health and development of control and experimental mice expressing different levels of ATM and APTX (*Figure 2*). We found that *Atm^R35X/R35X; Aptx^−/−* mice grew ~ 55% slower and reached estimated plateau weights that were ~ 35% less than control genotypes (log-rank, n=21–40, p<0.0001; *Figure 2A*). These differences in weight were a postnatal phenomenon, as no significant weight differences were detected just after birth (P8) across all genotypes (one-way

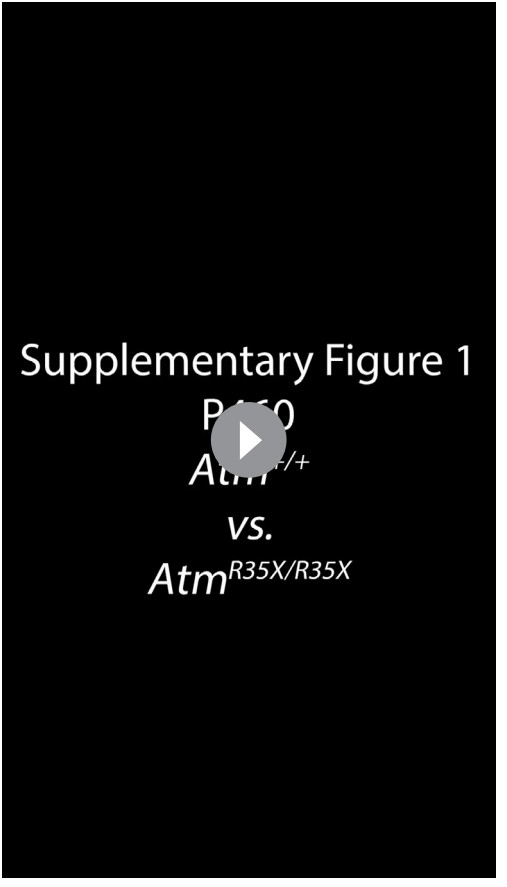

**Video 1.** Pole test—$Atm^{+/+}$ versus $Atm^{R35X/R35X}$; $Atm^{R35X/R35X}$ do not display an ataxic phenotype at P460.
https://elifesciences.org/articles/64695/figures#video1

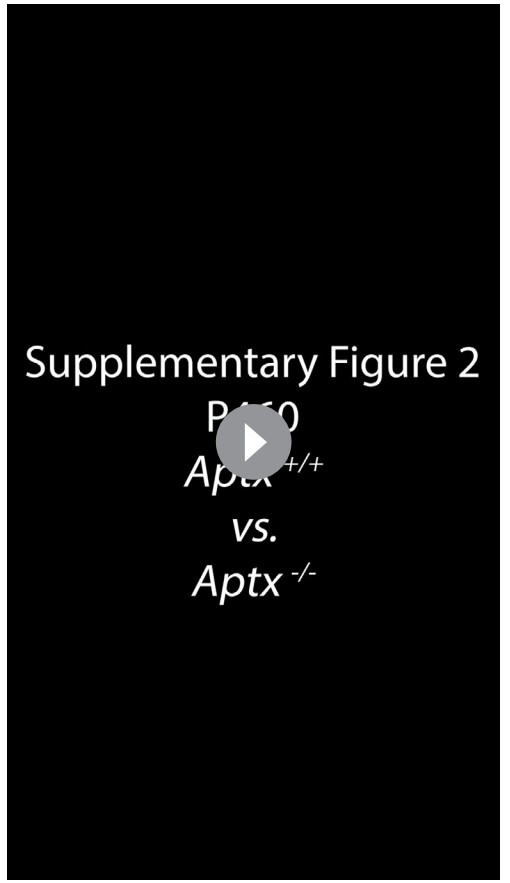

**Video 2.** Pole test—$Aptx^{+/+}$ versus $Aptx^{-/-}$; $Aptx^{-/-}$ mice do not display an ataxic phenotype at P460.
https://elifesciences.org/articles/64695/figures#video2

ANOVA, n=5–23, p>0.23). Adolescent double mutant mice at postnatal day 45 (P45) weighed on average 30% less in male mice (double mutant: 14.4±1.0 g [n=13] vs. wild-type: 20.2±0.5 g [n=16], t-test, p<0.0001) and 25% less in females (double mutant: 12.7±0.6 g [n=17] vs. wild-type: 17.0±0.2 g [n=15], t-test, p<0.0001; *Figure 1A*). Differences across the control genotypes were observed, but they were small and not consistent across time points or sex and therefore judged to not be physiologically relevant (*Figure 2A*). Survivability of the $Atm^{R35X/R35X}$; $Aptx^{-/-}$ mice was significantly reduced compared to $Atm^{+/+}$; $Aptx^{+/+}$ mice, with 53% of mice still alive at 400 days of age, compared to 97% of $Atm^{+/+}$; $Aptx^{+/+}$ mice at the same time point (*Figure 2B*). ATM deficiency alone was sufficient to reduce survivability; as compared to $Atm^{+/+}$; $Aptx^{+/+}$ mice, both $Atm^{R35X/R35X}$; $Aptx^{+/+}$ and $Atm^{R35X/R35X}$; $Aptx^{+/-}$ had significantly reduced survivability (42%, log-rank, $\chi^2_{(1, 56)}$=13.49, p=0.0002% and 52%, log-rank, $\chi^2_{(1, 53)}$=19.54, p<0.0001, respectively). No significant difference in survivability between ATM-deficient mice with partial or complete APTX deficiency was detected (log-rank, $\chi^2_{(2, 85)}$=1.01, p=0.6). Conversely, mice harboring at least one functional copy of the *Atm* gene had normal survivability, regardless of whether they expressed APTX or not (log-rank, $\chi^2_{(3, 131)}$=3.08, p=0.4). No significant difference between male and female mice was observed, and thus data were pooled (log-rank, p>0.4 for all pairwise comparisons; *Figure 2—figure supplement 2–1B*). Generally, a third of mice with ATM deficiency died from complications related to large thymic cancers (thymoma) found in the thoracic cavity (*Figure 2C*). The presence or absence of APTX did not impact cancer prevalence, and mice with at least one *Atm* transcript were cancer-free up until at least P400. Overall, ATM but not APTX deficiency had severe effects on the health and survivability of mice.

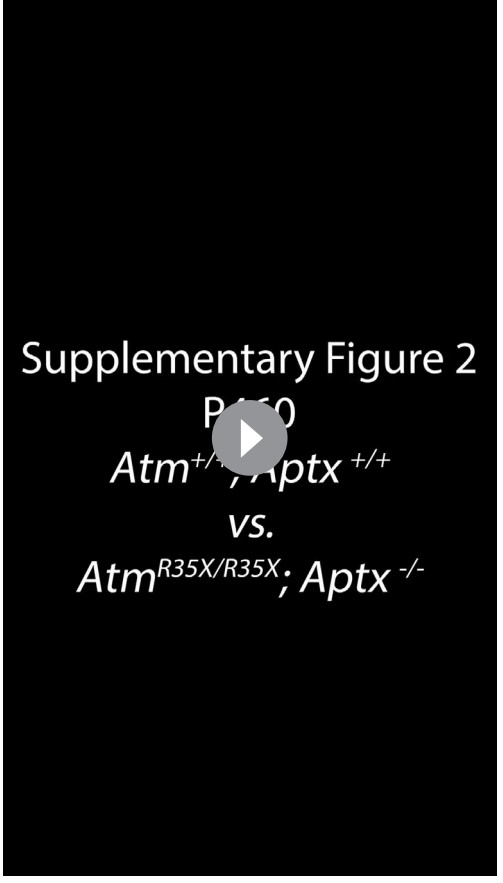

**Video 3.** Pole test—*Atm*<sup>+/+</sup>; *Aptx*<sup>+/+</sup> versus *Atm*<sup>R35X/R35X</sup>; *Aptx*<sup>−/−</sup>. *Atm*<sup>R35X/R35X</sup>; *Aptx*<sup>−/−</sup> display considerable motor disability at P460.
https://elifesciences.org/articles/64695/figures#video3

## Both ATM and APTX deficiency are necessary to produce progressive motor dysfunction

The progressive development of severe ataxia is a hallmark characteristic of A-T that is recapitulated in the $Atm^{R35X/R35X}$; $Aptx^{-/-}$ mice but none of the other control genotypes we tested. Overall, we find motor coordination deficits emerge between 210 and 400 days after birth in $Atm^{R35X/R35X}$; $Aptx^{-/-}$ mice and find no evidence of ataxia in mice with at least one copy of the $Atm$ or $Aptx$ gene (*Figure 3A and B*). For the vertical pole test, $Atm^{R35X/R35X}$; $Aptx^{-/-}$ mice took twice as long to descend at P400 compared to $Atm^{+/+}$; $Aptx^{+/+}$, $Atm^{+/+}$; $Aptx^{-/-}$, $Atm^{R35X/R35X}$; $Aptx^{+/+}$, or $Atm^{R35X/+}$; $Aptx^{-/-}$ mice (Male: 29.1 ± 0.9 s [n=3] vs. 7.5 ± 0.4 s [n=12], 12.5±2.5 s [n=9], 9.2±0.9 s [n=10], 8.6±0.9 s [n=11], one-way ANOVA, $F_{(4, 40)}$=19.9, p<0.0001; Female: 19.0±4.0 s [n=4] vs. 7.5±0.4 s [n=12], 7.8 ± 0.4 s [n=10], 10.5±1.2 s [n=6], 8.2±0.5 s [n = 8], one-way ANOVA, $F_{(4, 35)}$=13.9, p<0.0001). An examination of gait indicated that $Atm^{R35X/R35X}$; $Aptx^{-/-}$ mice at P400, but not P210 need additional stabilization during ambulation, as they spend twice as much time with three paws, rather than the normal 2, in contact with the ground as they walk across the gait analysis platform (Male: 56.2% vs. 26.4–32.2%, one-way ANOVA, $F_{(4, 54)}$=14.3, p<0.0001; Female: 58.4% vs. 18.9–28.8%, one-way ANOVA, $F_{(3, 178)}$=95.5, p<0.0001; *Figure 3B*). $Atm^{R35X/R35X}$; $Aptx^{-/-}$ also display a slower cadence and average speed across the platform compared to all other genotypes at P400 (cadence, Male: 9.5 vs. 13.3–15.9 steps/s, one-way ANOVA, $F_{(3, 204)}$=36.8, p<0.0001; Female: 9.1 vs. 14.2–15.9 steps/s, one-way ANOVA, $F_{(3, 204)}$=39.7, p<0.0001; speed, Male: 8.8 vs. 22–26 cm/s, one-way ANOVA, $F_{(4, 50)}$=28.3, p<0.0001; Female: 58.4 vs. 18.9–28.8 cm/s, one-way ANOVA, $F_{(3, 178)}$=39.7, p<0.0001; *Figure 3B*; *Figure 3— figure supplement 3–1*). This difference in speed and cadence is unlikely to be caused by animal size, as there are no significant differences in these parameters at earlier time points when the difference in size is significant (*Figure 2A*). These observations across the two behavioral tests were found in both male and female mice at each of their respective time points, consistent with the lack of sex differences observed in A-T patients.

We further examined behavioral differences between the $Atm^{R35X/R35X}$; $Aptx^{-/-}$ and $Atm^{+/+}$; $Aptx^{+/+}$ mice using a standardized set of experimental procedures used to phenotype genetically modified mice (i.e., SHIRPA; *Figure 3C*; *Figure 3—figure supplement 3–1*; *Rogers et al., 1997*). We first detected differences in motor function at P8, where $Atm^{R35X/R35X}$; $Aptx^{-/-}$ mice took 3–4 times longer on average to right themselves compared to $Atm^{+/+}$; $Aptx^{+/+}$ mice (Male: 6.4±1.1 s [n=24] vs. 1.5±0.1 s [n=23], t-test, p<0.0002; Female: 11.1±1.9 s [n=21] vs. 2.4±0.3 s [n=17], t-test, p<0.0002; *Figure 3C* bottom). At

**Video 4.** Open field—*Atm*<sup>+/+</sup>; *Aptx*<sup>+/+</sup> versus *Atm*<sup>R35X/R35X</sup>; *Aptx*<sup>−/−</sup>. *Atm*<sup>R35X/R35X</sup>; *Aptx*<sup>−/−</sup> display a clear inability to ambulate in the open field at P460.
https://elifesciences.org/articles/64695/figures#video4

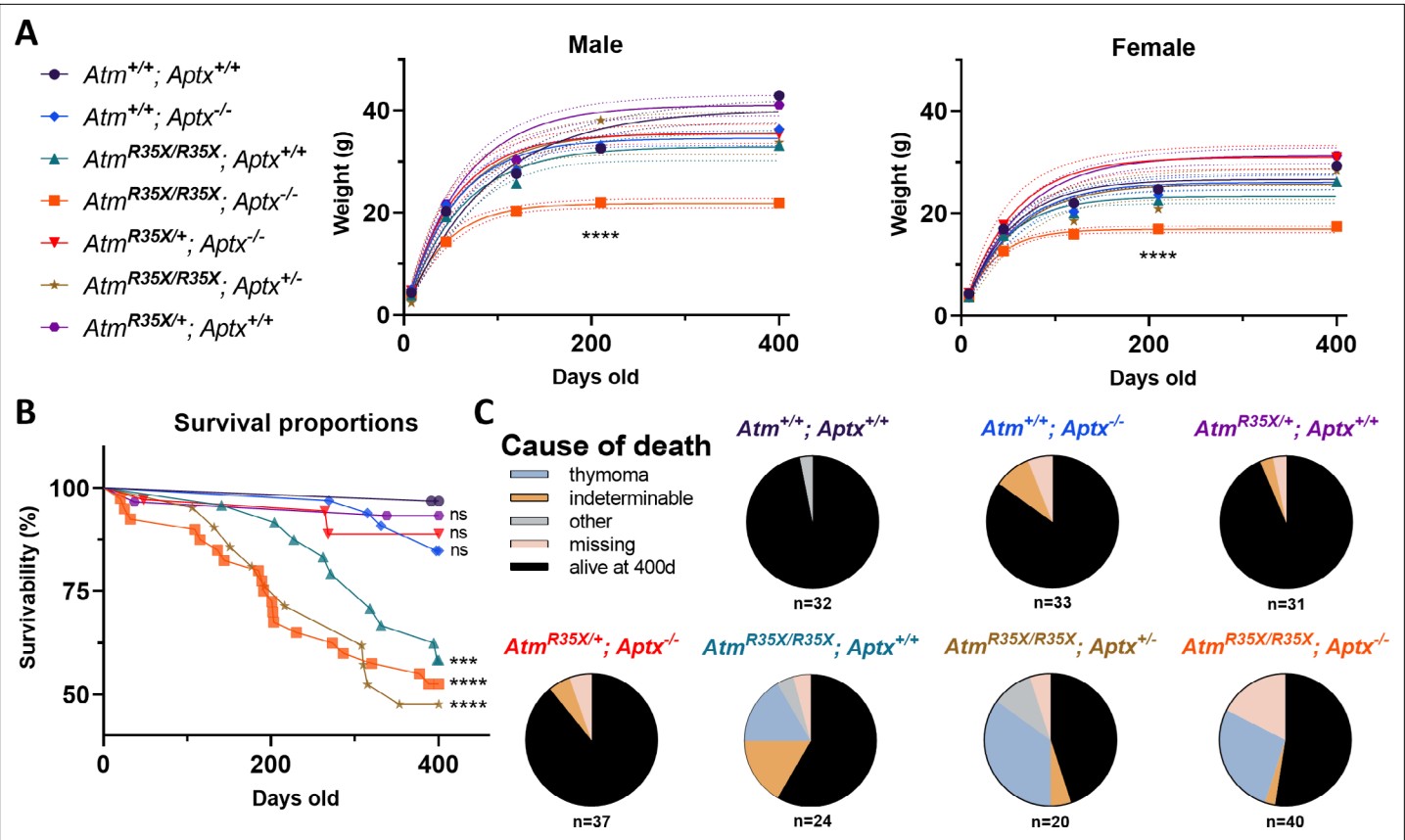

**Figure 2.** Health and survivability of single and double mutant mice. (**A**) *Left*: The line color and symbol for each genotype are denoted and are consistent across all figures. *Right*: $Atm^{R35X/R35X}$; $Aptx^{-/-}$ mice weighed significantly less than all control genotypes as indicated by the growth curves (± 95% confidence interval; dotted lines). Growth curve ($Atm^{R35X/R35X}$; $Aptx^{-/-}$ vs. controls): Male k=0.024 vs 0.011–0.019, $Y_{max}$=21.8 vs. 32.9–41.0 g (n=3–18); Female k=0.030 vs. 0.017–0.022, $Y_{max}$=16.9 vs 23.3–31.3 (n=2–19). Sum of squares F-test run across all curves: Male $F_{(12, 364)}$=30.5, p<0.0001, Female $F_{(12, 339)}$=28.3, p<0.0001. (**B**) ATM-deficient mice, regardless of APTX expression, displayed significantly lower survivability with ~ 55% of mice deceased by P400. No statistical differences between ATM-deficient mice were detected. Moreover, a single wild-type copy of the *Atm* gene was sufficient to prevent premature death (no statistical difference detected between $Atm^{R35X/+}$; $Aptx^{-/-}$ and $Atm^{+/+}$; $Aptx^{+/+}$ mice). Log-rank (Mantel-Cox) tests across all ($\chi^2_{(6,217)}$=48.4, p<0.0001), just the ATM-deficient ($\chi^2_{(2,217)}$=1.06, p=0.6), and single comparisons to wild-type (see figure) were conducted. Total number of animals indicated in panel (**C**). (**C**) Pie charts illustrating that ATM-deficient mice displayed a high prevalence of thymomas based on postmortem necropsies. 'Other' probable causes of death included enlarged livers and obstructed kidneys. 'Missing' mice were presumed dead and cannibalized by cage mates, cause of death unknown. ATM, Ataxia Telangiectasia mutated.

The online version of this article includes the following source data and figure supplement(s) for figure 2:

**Source data 1.** Numerical data of weight, age of death, and probable cause of death.

**Figure supplement 1.** Animal weight for each time point and genotype.

30 days of age, we detected significant differences between $Atm^{R35X/R35X}$; $Aptx^{-/-}$ and $Atm^{+/+}$; $Aptx^{+/+}$ mice in behavioral tests that qualitatively measure body position and spontaneous activity (*Figure 3C*). Striking differences in $Atm^{R35X/R35X}$; $Aptx^{-/-}$ compared to $Atm^{+/+}$; $Aptx^{+/+}$ mice were observed at P400, especially for behaviors related to movement, including locomotor activity, body position, and gait (*Figure 3C*). The results from this battery of tests demonstrate that $Atm^{R35X/R35X}$; $Aptx^{-/-}$ mice develop a severe change in behavior by P400, consistent with purely visual observations of significant motor coordination deficits in the mice up to this time point. Importantly, we do not find any significant differences between the other control genotypes, including $Atm^{R35X/+}$; $Aptx^{-/-}$ mice that express at least some ATM but no APTX protein (*Figure 3—figure supplement 3–1*).

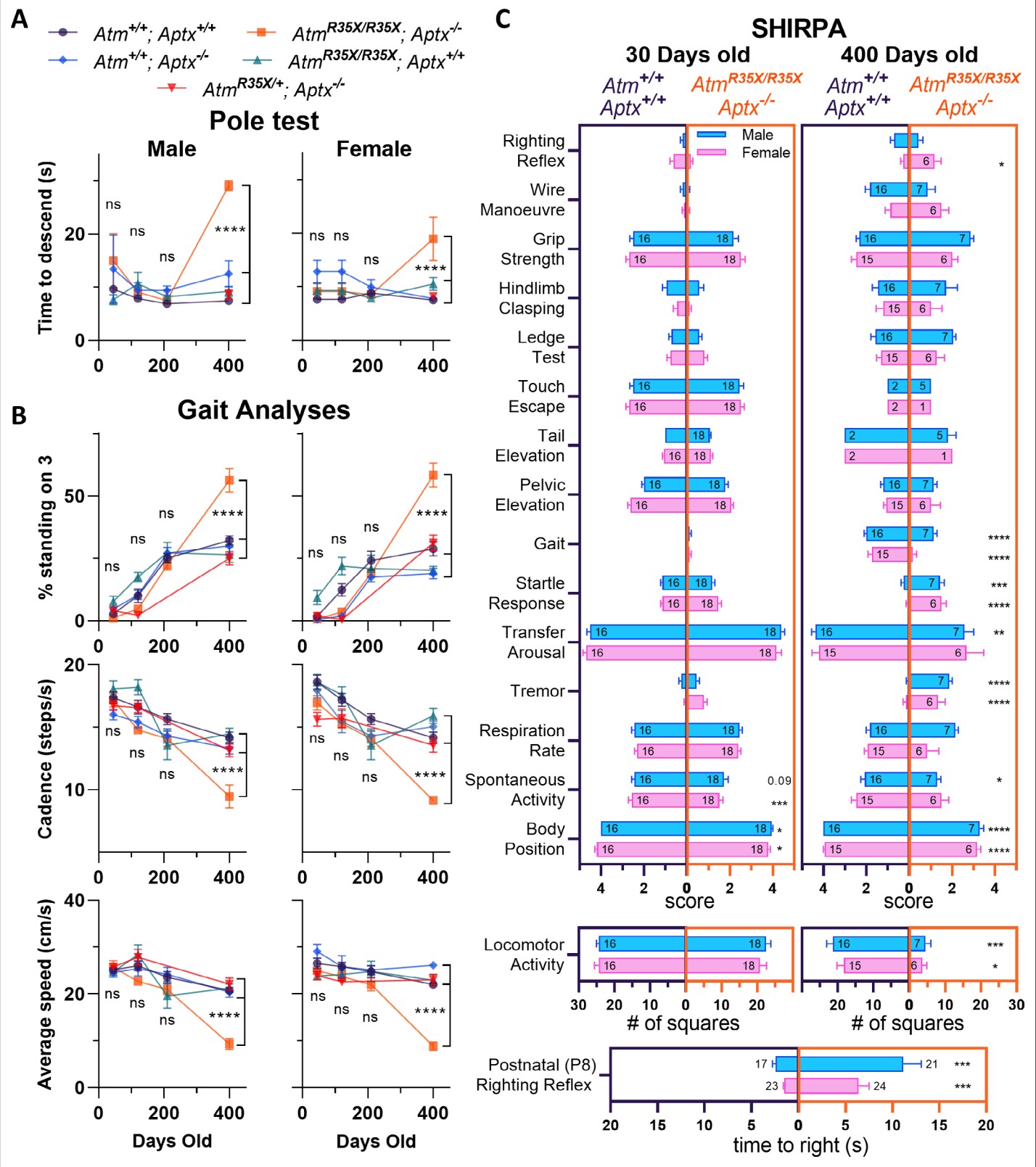

**Figure 3.** $Atm^{R35X/R35X}$; $Aptx^{-/-}$ mice develop a progressive loss in motor coordination. (**A**) $Atm^{R35X/R35X}$; $Aptx^{-/-}$ mice take a similar amount of time to descend a vertical pole at P45, P120, and P210, but significantly longer at P400. These overall results were found to be similar for both male (left, n=2–12) and female (right, n=4–12) mice. (**B**) Consistent with the vertical pole test, the gait of $Atm^{R35X/R35X}$; $Aptx^{-/-}$ mice measured during ambulation on a Catwalk gait analysis system was significantly different to controls by P400, but not before P210. This includes the percent of time a mouse spends

*Figure 3 continued*

with 3 versus 1, 2, or 4 paws on the ground and the speed and cadence during each run across the platform. The effects of the two null mutations were generally similar between males (left, n=4–21) and females (right, n=3–18). (**C**) Behavioral data for male (blue) and female (pink) *Atm^{R35X/R35X}; Aptx^{−/−}* (dark purple, left) and *Atm^{+/+}; Aptx^{+/+}* (orange, right) mice are plotted at P30 (left two columns) and P400 (right two columns). Left-right asymmetries of the horizontal bars indicate a difference in performance between genotypes for the behavioral test listed in a column on the far left. A significant difference in the time to right during the righting reflex at P8 was observed in both male and female mice (bottom). (**A**) and (**B**) were examined via two-way ANOVA with age and genotype as factors followed by Tukey's multiple comparison tests between *Atm^{R35X/R35X}; Aptx^{−/−}* and each of the control genotypes. Behavioral tests in (**C**) were examined using a non-parametric Kruskal-Wallis followed by Dunn's multiple comparisons tests. Symbol/color key: *Atm^{+/+}; Aptx^{+/+}* (purple circle), *Atm^{+/+}; Aptx^{−/−}* (blue diamond), *Atm^{R35X/R35X}; Aptx^{+/+}* (green triangle), *Atm^{R35X/R35X}; Aptx^{−/−}* (orange square), *Atm^{R35X/+}; Aptx^{−/−}* (red inverted triangle).

The online version of this article includes the following figure supplement(s) for figure 3:

**Source data 1.** Numerical data for animal behavioral assessments.

**Figure supplement 1.** *Atm^{R35X/R35X}; Aptx^{−/−}* mice develop progressive ataxia.

## Membrane and synaptic properties are perturbed in ATM- and APTX-deficient Purkinje neurons

Purkinje neurons (PNs) are a key neuronal subtype located in the cerebellar cortex. They display considerable intrinsic excitability, firing action potentials (APs) spontaneously at rates significantly higher than most other neurons in the brain (50–100 Hz more in many cases). Their activity shapes cerebellar output via tonic inhibition of neurons of the cerebellar nuclei, which project to motor coordination centers in the forebrain, brainstem, and spinal cord. Cerebellar PN dysfunction is associated with several forms of ataxia and implicated in A-T (*Hoxha et al., 2018*; *Cook et al., 2021*; *Shiloh, 2020*). We therefore examined if the electrophysiological properties of PNs in the *Atm^{R35X/R35X}; Aptx^{−/−}* cerebellum were abnormal.

Since PN baseline activity and responsivity to input is mediated by a baseline set of passive and active membrane properties (*Figure 4*), we directly recorded from and compared the membrane properties of PNs in acute cerebellar sections harvested from *Atm^{R35X/R35X}; Aptx^{−/−}* and *Atm^{+/+}; Aptx^{+/+}* mice (P350–P400). PNs recorded from *Atm^{R35X/R35X}; Aptx^{−/−}* mice had significantly 'tighter' membranes, displaying higher membrane input resistances ($R_m$) than those from *Atm^{+/+}; Aptx^{+/+}* mice (47.7±5.6 [n=15] vs. 30.2±1.47 [n=23] MΩ, t-test, p=0.008; *Figure 4B*) and displayed a faster membrane time constant ($\tau$) (3.6±0.4 [n=15] vs. 5.1±0.3 [n=23] ms, t-test, p=0.009; *Figure 4B*). These results indicate that the total membrane capacitance ($C_m = \tau/R_m$) of the *Atm^{R35X/R35X}; Aptx^{−/−}* PNs is significantly reduced (98.25±19.23 [n=15] vs. 175.6±12.67 [n=23] pF, t-test, p=0.0025; *Figure 4B*). At the cellular level, this suggests that the ATM- and APTX-deficient PNs have less (i.e., decreased area) or thinner membranes than that of wild-type PNs, a result suggestive of a developmental deficit or neurodegenerative process (*Dell'Orco et al., 2015*). We next assessed the intrinsic excitability of PNs in *Atm^{R35X/R35X}; Aptx^{−/−}* and *Atm^{+/+}; Aptx^{+/+}* mice by examining PN AP generation and dynamics. Significant deficits in the ability of PNs to fire continuously in response to current injection were observed in *Atm^{R35X/R35X}; Aptx^{−/−}* mice (*Figure 4C*). These deficits were associated with significant perturbations in the amplitude, threshold, and area of evoked APs (amplitude: 66.2±0.7 [n=14] vs. 72.1±1.4 [n=13] ΔmV, t-test, p=0.003; threshold: −55.2±1.5 to −48.61±1.9 mV, t-test, p=0.0196; area: 17.96±0.6 vs. 20.63±1.0 mV*ms, t-test, p=0.048; *Figure 4C*). Taken together, these experiments demonstrate significant perturbations of PN physiological properties that likely disrupt their ability to function normally in the cerebellum of *Atm^{R35X/R35X}; Aptx^{−/−}* mice.

We next tested whether extrinsic and/or synaptic PN properties were also impacted in *Atm^{R35X/R35X}; Aptx^{−/−}* mice. We first examined spontaneous excitatory postsynaptic currents (sEPSCs) generated by granule cell-to-PN synapses (i.e., parallel fiber inputs). No difference in sEPSC size was detected, indicating the function of granule cell axon terminals (i.e., parallel fibers) was relatively normal in the *Atm^{R35X/R35X}; Aptx^{−/−}* cerebellum (18.92±1.3 [n=11] vs. 23.4±3.3 [n=11] pA, t-test, p=0.477; *Figure 4D*, *Yamasaki et al., 2006*). sEPSC frequency, however, was found to be significantly increased, a phenomenon that could be attributed to either an increase in the total number of synapses, an increase in the size of the readily releasable pool of synaptic vesicles, or an increase in the probability of neurotransmitter release in PNs of *Atm^{R35X/R35X}; Aptx^{−/−}* mice (18.75±2.8 Hz [n=11] vs. 11.4±1.0 Hz [n=11], t-test, p=0.047; *Figure 4D*). We next explored evoked synaptic release and short-term plasticity by simultaneously recording from PNs and electrically stimulating either granule cell (i.e., parallel fibers) or

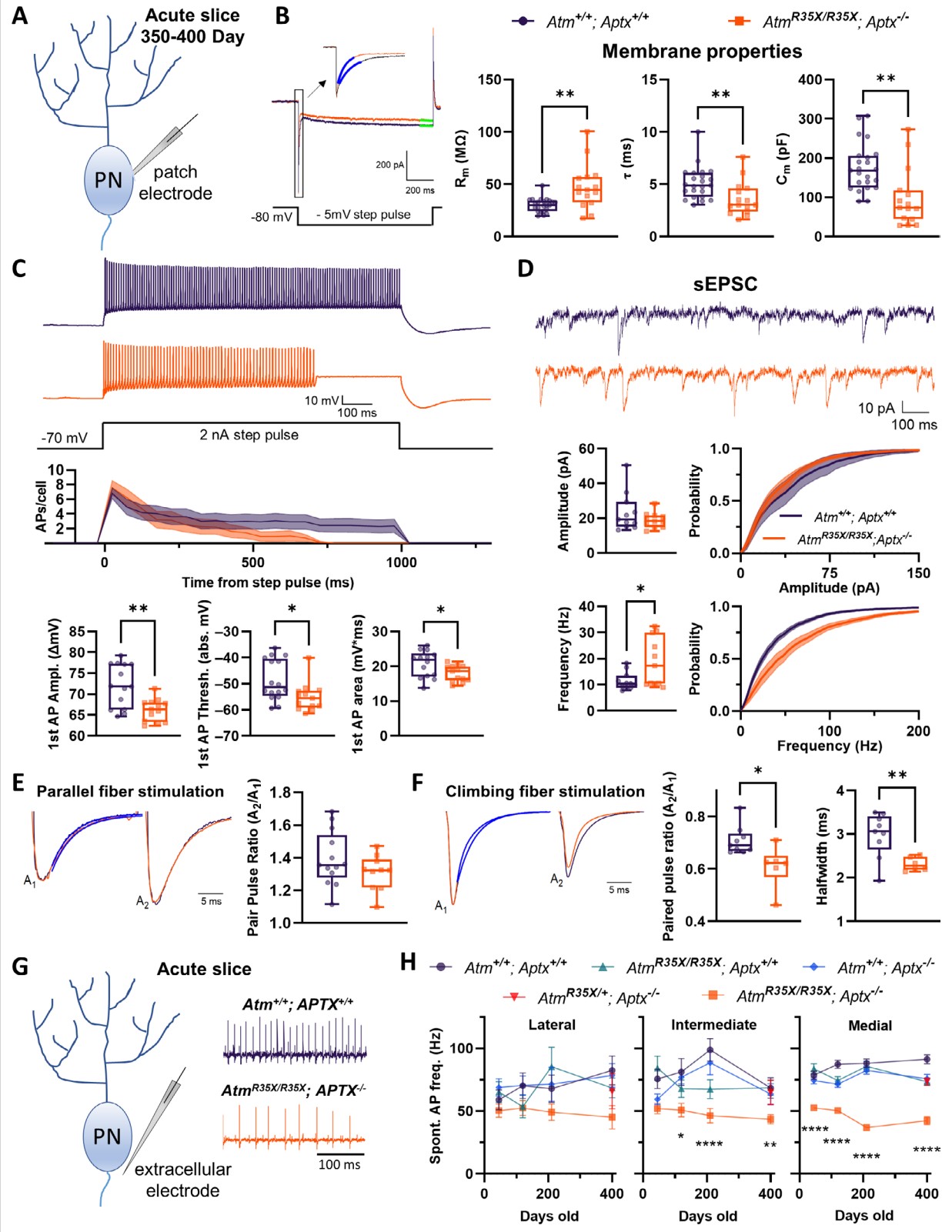

**Figure 4.** The biophysical properties of PNs are significantly perturbed in $Atm^{R35X/R35X}$; $Aptx^{-/-}$ mice. (**A**) Schematic diagram of intracellular recording from a single Purkinje neuron (PN) in an acute cerebellar tissue slice preparation used to examine their physiological properties. (**B**) *Left:* Voltage-clamp measurements of PN membrane properties made from a 1 s, –5 mV step pulse as illustrated. *Right:* The membrane input resistance ($R_m$), time constant ($\tau$), and capacitance ($C_m$) were perturbed in $Atm^{R35X/R35X}$; $Aptx^{-/-}$ compared to $Atm^{+/+}$; $Aptx^{+/+}$ mice. (**C**) Current-clamp recordings of PN action potentials

*Figure 4 continued on next page*

*Figure 4 continued*

(APs) after 2 nA step pulses from a –70 mV holding potential. PN APs recorded from *Atm^R35X/R35X^; Aptx^–/–^* fail to maintain constant firing and summary plots show that they have lower 1st AP amplitudes, firing threshold, and area under the curve. (**D**) *Top:* Example sEPSC traces taken from a PN under voltage clamp at a –80 mV holding potential. *Bottom:* Median frequency and amplitude data, along with the overall probability distribution function are plotted for both *Atm^+/+^; Aptx^+/+^* (n=11) and *Atm^R35X/R35X^; Aptx^–/–^* (n=11) mice. The frequency but not amplitude of PNs recorded in *Atm^R35X/R35X^; Aptx^–/–^* mice was found to be perturbed. (**E, F**) *Left:* Example traces of evoked EPSCs recorded from PNs as a result of a two-pulse stimulation (50 ms interval) of either parallel (**E**) or climbing (**F**) fiber axons. Traces illustrate the first (A₁) and second (A₂) amplitude (normalized) and time course of decay (blue fitted line) of each synaptic response. *Right:* Summary plots of the paired-pulse ratio. While parallel fiber paired-pulse facilitation was normal in *Atm^R35X/R35X^; Aptx^–/–^* mice, climbing fiber paired-pulse depression and halfwidth was significantly perturbed compared to *Atm^+/+^; Aptx^+/+^* mice. (**G**) Schematic diagram of extracellular recording from a single PN in an acute cerebellar tissue slice preparation. Example electrophysiological traces for *Atm^+/+^; Aptx^+/+^* (purple, top) and *Atm^R35X/R35X^; Aptx^–/–^* (orange, bottom) PNs in the medial area (i.e., vermis) of the cerebellum. (**H**) *Atm^R35X/R35X^; Aptx^–/–^* PN AP firing frequency progressively decreased with age and was significantly slower in comparison to all control genotypes expressing at least one copy of the *Atm* or *Aptx* gene [*Atm^+/+^; Aptx^+/+^* (n=52–59), *Atm^+/+^; Aptx^–/–^* (n=51–64), *Atm^R35X/R35X^; Aptx^+/+^* (n=39–52), *Atm^R35X/R35X^; Aptx^–/–^* (n=24–71), *Atm^R35X/+^; Aptx^–/–^* (n=69)]. Data in (**B**) were compared using an ANOVA (Kruskal-Wallis) followed by Dunn's multiple comparisons test, data in (**D–F**) were compared via Welch's t-test, and data in (**H**) using a two-way ANOVA followed by Holm-Šídák's multiple comparisons test. Symbol/color key: *Atm^+/+^; Aptx^+/+^* (purple circle), *Atm^+/+^; Aptx^–/–^* (blue diamond), *Atm^R35X/R35X^; Aptx^+/+^* (green triangle), *Atm^R35X/R35X^; Aptx^–/–^* (orange square), *Atm^R35X/+^; Aptx^–/–^* (red inverted triangle). sEPSC, spontaneous excitatory postsynaptic current.

The online version of this article includes the following source data and figure supplement(s) for figure 4:

**Source data 1.** Numerical data of electrophysiological recordings for each panel.

**Figure supplement 1.** Current versus voltage responses significantly differ between *Atm^+/+^; Aptx^+/+^* and *Atm^R35X/R35X^; Aptx^–/–^* mice.

**Figure supplement 2.** Mean PN firing frequency across the cerebellum.

**Figure supplement 3.** Mean PN firing frequency across genotype and sex.

**Figure supplement 4.** Coefficient of variation (CV) of PN firing frequency across the cerebellum.

**Figure supplement 5.** Mean variation between PN firing intervals across the cerebellum.

---

inferior olivary (i.e., climbing fiber) axons with a paired-pulse burst (two-pulses, 50 ms apart). The synaptic properties of parallel fibers were found to be normal, displaying no significant differences in the expected short-term facilitation (*Atluri and Regehr, 1996*) or halfwidth and decay time constant of the evoked EPSC (PPR: 1.3±0.03 [n=10] vs. 1.4±0.05 [n=13], t-test, p=0.162; halfwidth: 3.9±0.6 vs. 4.9±0.4 ms, t-test, p=0.175; time constant: 3.5±0.5 vs. 4.7±0.4 ms, t-test, p=0.054; *Figure 4E*). In comparison, we found climbing fiber-to-PN synaptic responses, which normally display pair-pulse depression (*Hansel and Linden, 2000*), to depress at significantly greater magnitudes in *Atm^R35X/R35X^; Aptx^–/–^* mice (PPR: 0.6±0.03 [n=6] vs. 0.7±0.02 [n=9], t-test, p=0.03; *Figure 4F*). The overall width and decay time constant of the evoked currents were also smaller (halfwidth: 2.3±0.6 [n=6] vs. 3.0±0.2 [n=9] ms, t-test, p=0.004; time constant (fast): 1.1±0.14 vs. 2.9±0.4 ms, t-test, [p=0.001]). While these results could be caused by a presynaptic deficit, such as reduced vesicle stores in the climbing fiber axon terminal, the unaffected initial magnitude of the EPSC (2.4±0.4 [n=6] vs. 1.9±0.2 nA [n=9], t-test, p=0.3) points to a more intrinsic deficit, such as a reduced $Ca^{2+}$ influx from the endoplasmic reticulum, which could significantly impact long-term synaptic plasticity critical to cerebellar function (*Hoxha et al., 2018*; *Kano and Watanabe, 2017*). Overall, the perturbations to the passive and active PN properties that we observe here likely give rise to the significant cerebellar dysfunction in *Atm^R35X/R35X^; Aptx^–/–^* mice.

## ATM and APTX deficiency causes a progressive perturbation of PN neural activity that is associated with dendritic shrinking and overall cerebellar atrophy

Decreased rates of spontaneous PN AP firing, which can be indicative of PN dysfunction, have been observed in several mouse models of ataxia, including spinocerebellar ataxias (SCA) 2, 3, 5, 6, 13, and 27, several models of episodic ataxia (e.g., leaner, ducky, and tottering), and autosomal-recessive spastic ataxia of the Charlevoix-Saguenay (*Hourez et al., 2011*; *Hansen et al., 2013*; *Dell'Orco et al., 2017*; *Kasumu and Bezprozvanny, 2012*; *Liu et al., 2009*; *Perkins et al., 2010*; *Shakkottai et al., 2011*; *Jayabal et al., 2016*; *Stoyas et al., 2020*; *Hurlock et al., 2008*; *Shakkottai et al., 2009*; *Bosch et al., 2015*; *Walter et al., 2006*; *Alviña and Khodakhah, 2010*; *Ady et al., 2018*; *Larivière et al., 2019*; *Cook et al., 2021*). We therefore used this biomarker to characterize the progression of PN

perturbation in $Atm^{R35X/R35X}$; $Aptx^{-/-}$ mice and assess whether deficits were restricted to ATM- and APTX-deficient mice, consistent with the behavioral results (*Figures 2 and 3*). We additionally examined whether decreased PN activity differed across the cerebellum, as anecdotal clinical pathology reports suggest degeneration may occur asymmetrically across the cerebellum, with the anterior and posterior vermis and middle cerebellar hemispheres affected the most, although no systematic analysis has been performed, and the consistency of results across patients is highly variable (*Verhagen et al., 2012*; *De León et al., 1976*; *Amromin et al., 1979*; *Monaco et al., 1988*; *Terplan and Krauss, 1969*; *Strich, 1966*; *Solitare, 1968*; *Solitare and Lopez, 1967*; *Aguilar et al., 1968*; *Paula-Barbosa et al., 1983*).

Using extracellular recording methods in the acute slice, we recorded spontaneous APs from 3300 PNs (*Figure 4G*) across 188 animals, encompassing $Atm^{R35X/R35X}$; $Aptx^{-/-}$ and three other genotypes at four different time points (P45, P120, P210, and P400). We visually selected 'healthy' cells (see Materials and methods) located deeper in the slice, that consistently fired during the extent of the 60-s recording period. Qualitatively, tissue and cell quality did not visually differ across genotypes under DIC microscopy. Cells were sampled in a distributed fashion across the lateral, intermediate, and medial (vermis) cerebellum of each mouse to assess whether changes in PN firing activity were ubiquitous or anatomically restricted. Regions were segregated based on gross anatomical domains in the mouse defined by natural anatomical boundaries (e.g., foliation) and their general connectivity with different regions of the nervous system (e.g., forebrain, brainstem, etc.) (*Voogd and Glickstein, 1998*).

We found that complete deficiency of both ATM and APTX, consistent with the behavioral results, was necessary to produce a significantly reduced spontaneous PN firing frequency (*Figure 4G and H*). Although the trend of slower PN firing rates was observed across most regions of the cerebellum, some subregions appeared to be less or minimally impacted, including several areas of the lateral cerebellum, including the paraflocculus, paramedian, and crus I and II (*Figure 4—figure supplement 4–2*). Significant age-dependent changes in firing frequency were also only observed in $Atm^{R35X/R35X}$; $Aptx^{-/-}$ mice (*Figure 4H*), with the most significant decline occurring between P120 and P210 (medial: 50.3±2.4 Hz [n=61] vs. 36.9±2.2 Hz [n=31], t-test, p=0.0006). No significant difference in PN firing frequency was detected between male and female mice within each genotype, thus the data were pooled (two-way ANOVA, p>0.3 across all pairwise comparisons; *Figure 4—figure supplement 4–3*). Previous studies across several mouse models of heritable ataxia, including episodic ataxia and several variants of spinocerebellar ataxia find that physiological disruption in PN firing not only changes its frequency, but also its regularity (*Kasumu and Bezprozvanny, 2012*; *Jayabal et al., 2016*; *Stoyas et al., 2020*; *Cook et al., 2021*). We compared both the coefficient of variation (CV) and variability in adjacent intervals (CV2) between $Atm^{R35X/R35X}$; $Aptx^{-/-}$ and control mice (*Figure 4—figure supplements 4 and 5*). No difference in these parameters across sex, age, or genotype was detected. Consistent with the behavioral results, cerebellar dysfunction was found only in the $Atm^{R35X/R35X}$; $Aptx^{-/-}$ mice that developed ataxia and not in mice with partial or full expression of ATM or APTX.

## ATM-and APTX-deficiency induces cerebellar atrophy

In A-T patients, ataxia is usually detected between 1 and 2 years of age and is associated with little to no cerebellar atrophy (*Tavani et al., 2003*; *Taylor et al., 2015*). Significant structural changes and atrophy are usually first detected via neuroimaging between 5 and 10 years of age (*Demaerel et al., 1992*; *Tavani et al., 2003*). Postmortem clinical histopathology in A-T patients points to significant changes in PN morphology and density; however, these reports primarily detail patients at late stages of the disorder, and the relationship between the severity of PN pathology and ataxia is not clear (*Verhagen et al., 2012*; *De León et al., 1976*; *Amromin et al., 1979*; *Monaco et al., 1988*; *Terplan and Krauss, 1969*; *Strich, 1966*; *Solitare, 1968*; *Solitare and Lopez, 1967*; *Aguilar et al., 1968*; *Paula-Barbosa et al., 1983*; *Gatti and Vinters, 1985*).

In the $Atm^{R35X/R35X}$; $Aptx^{-/-}$ mice, we found the gross size of the cerebellum to be normal early in life, but significant atrophy developed as the severity of ataxia increased (*Figure 5A*). At early stages (P45–P210), the size of the cerebellum in $Atm^{R35X/R35X}$; $Aptx^{-/-}$ mice did not differ from mice with at least one copy of the *Atm* gene (two-way ANOVA, $F_{(3,52)}=1.0$, p=0.4). However, by P210, the overall size of the cerebellum in $Atm^{R35X/R35X}$; $Aptx^{-/-}$ mice was significantly reduced (one-way ANOVA, $F_{(3,37)}=1.4$, p=0.3), with the degenerative process continuing to at least the last time point investigated (P460).

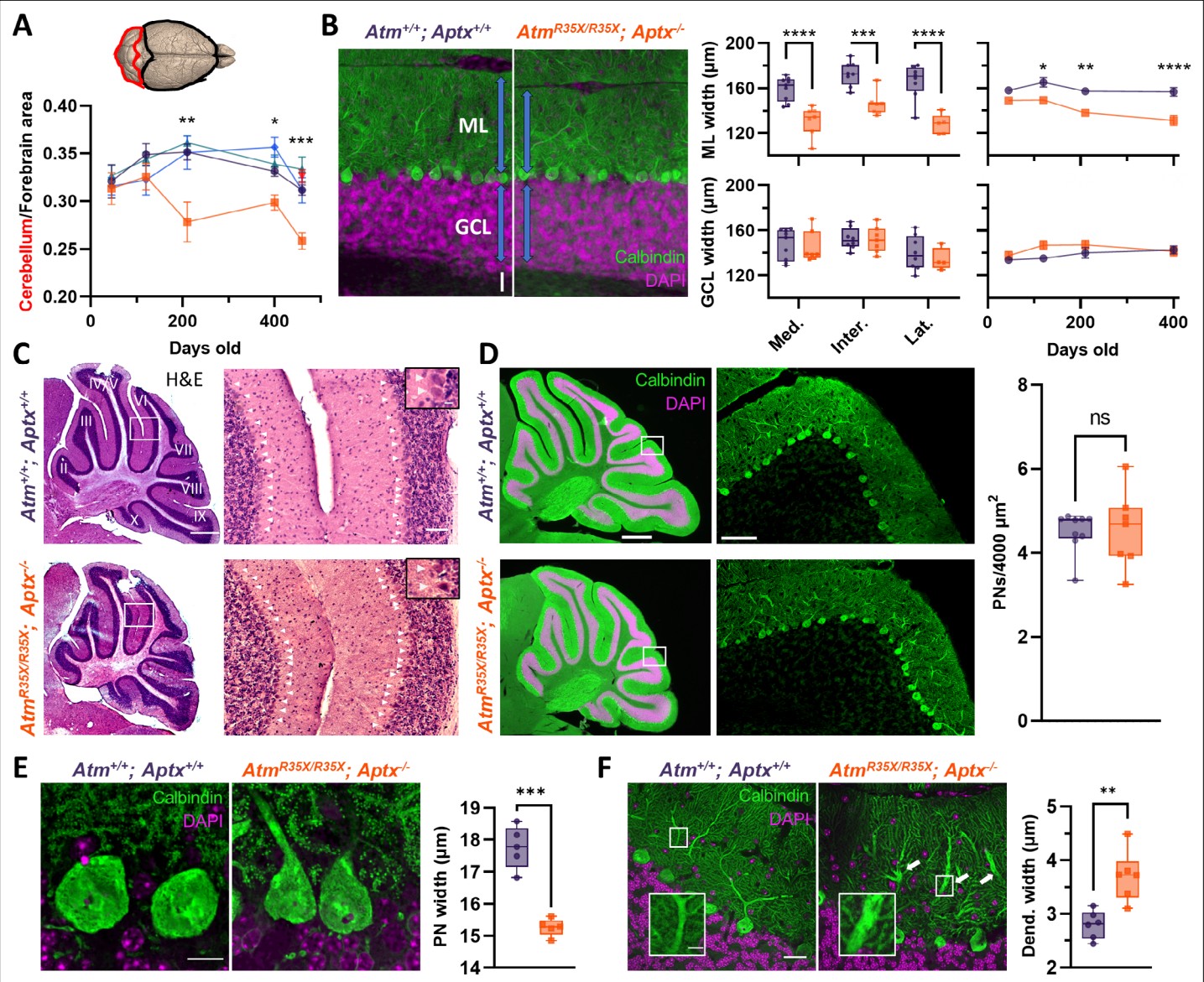

**Figure 5.** Cerebellar atrophy is associated with a progressive reduction in molecular layer (ML) width and pathological changes in PN morphology but not PN cell death. (**A**) Cartoon image of the brain highlighting the dorsal forebrain and cerebellar surface. Area estimates from dorsal images of the brain were used to determine the cerebellum to forebrain ratio allowing us to control for any differences in overall size of the brain. We found the cerebellum decreased in size over age in $Atm^{R35X/R35X}$; $Aptx^{-/-}$ (n=5–10), but not control mice ($Atm^{+/+}$; $Aptx^{+/+}$ [n=4–20], $Atm^{+/+}$; $Aptx^{-/-}$ [n=4–12], $Atm^{R35X/R35X}$; $Aptx^{+/+}$ [n=6–16], $Atm^{R35X/+}$; $Aptx^{-/-}$ [n=6]). (**B**) *Left:* Immunofluorescent images of parasagittal cerebellar sections illustrating the length (blue arrows) of PN dendrites in the ML (green) and width of the granule cell layer (GCL; magenta) in $Atm^{+/+}$; $Aptx^{+/+}$ and $Atm^{R35X/R35X}$; $Aptx^{-/-}$ mice (P400; medial cerebellar lobule VIII). Scale bar=50 µm. *Middle:* Summary graphs of ML and GCL width measurements averaged across all lobules in sections from the medial, intermediate, and lateral parts of the cerebellum (P400). *Right:* Average ML and GCL widths at different ages in the medial cerebellum (n=5–7). (**C**) *Left:* H&E stained, parasagittal cerebellar sections from P400 $Atm^{+/+}$; $Aptx^{+/+}$ (top) and $Atm^{R35X/R35X}$; $Aptx^{-/-}$ (bottom) mice. Scale bar=500 µm *Right:* Magnified view (from white box) of PNs (white triangles) in cerebellar lobules 4/5 and 6. Scale bar=50 µm, inset 10 µm Right: The average density of PNs across all lobules in the medial cerebellum of $Atm^{+/+}$; $Aptx^{+/+}$ (n=9) and $Atm^{R35X/R35X}$; $Aptx^{-/-}$ (n=7) P400 mice do not differ. (**D**) *Left:* Whole cerebellar, fluorescent images and magnified view of folia VIII (from white box). Scale bar=500 µm left, 75 µm right. *Right:* Average density of Calbindin positive PNs across the whole cerebellum (n=7–9). (**E**) *Left:* Optically sectioned fluorescent images illustrate the smaller size of PNs in $Atm^{R35X/R35X}$; $Aptx^{-/-}$ mice. *Right:* Plot of the average width of PN somas randomly sampled from across the cerebellum (n=5). Scale bar=10 µm. (**F**) *Left:* Images like in (**E**). Abnormally large caliber PN dendrites (inset) were observed in $Atm^{R35X/R35X}$; $Aptx^{-/-}$ mice (P400; medial cerebellar lobule VI). Scale bar=25 µm main, 5 µm inset. *Right:* Average width of primary and secondary PN dendrites measured at the midline between the PN cell bodies and ML edge in $Atm^{+/+}$; $Aptx^{+/+}$ and $Atm^{R35X/R35X}$; $Aptx^{-/-}$ mice (n=6). Statistical significances were assessed via two-way ANOVA with age and genotype as factors followed by Holm-Šídák (**A** and **B** *right*) or Šídák (**B** *middle*) pairwise multiple comparisons test. Welch's t-test used in (**D–F**). PN, Purkinje neuron.

*Figure 5 continued on next page*

*Figure 5 continued*

The online version of this article includes the following figure supplement(s) for figure 5:

**Source data 1.** Numerical data for histology.

**Figure supplement 1.** Decreased molecular layer (ML) width but not cell death is a key feature of the A-T model.

**Figure supplement 2.** Cerebellar degeneration in the A-T model is not associated with micro glial activation or cell death markers but is associated with significant swelling of PN dendrites.

---

To rule out the possibility that reduced cerebellar size was related to the smaller stature of $Atm^{R35X/R35X}$; $Aptx^{-/-}$ mice, we examined, animal weight and cerebellar size and found no correlation (Pearson's correlation, p>0.3 for all four genotypes at P460, n=10–20). Furthermore, we found that cerebellar size did not differ between male and the on average 22% smaller female mice across genotypes at this age (two-way ANOVA, $F_{(2, 153)}$=1.9, p=0.2). Therefore, cerebellar neurodegeneration in the $Atm^{R35X/R35X}$; $Aptx^{-/-}$ mice, which begins after P120, is correlated with ATM and APTX deficiency.

We found cerebellar atrophy to be associated with a selective reduction in the width of the molecular layer (ML) where PN dendrites reside (*Figure 5B*). Consistent with the temporal changes in gross cerebellar size, PN firing frequency, and behavior, ML width in $Atm^{R35X/R35X}$; $Aptx^{-/-}$ mice was normal in younger mice but progressively decreased in width as the severity of ataxia increased (P400: 120.2±2.1 µm [n=5] vs. 140.2±4.8 µm [n=5], two-way ANOVA, $F_{(1, 42)}$=45.04, p<0.0001; *Figure 5B and Figure 5—figure supplement 1*). In contrast, no difference in the width of the granule cell layer (GCL) across age was observed in the $Atm^{R35X/R35X}$; $Aptx^{-/-}$ mice (P400: 135.5±2.4 µm [n=6] vs. 127.5±4.3 µm [n=5], two-way ANOVA, $F_{(1, 42)}$=3.3, p=0.08; *Figure 5B and Figure 5—figure supplement 1*). Cerebellar atrophy, however, was not due to PN cell death, as PN density did not significantly differ between $Atm^{R35X/R35X}$; $Aptx^{-/-}$ and $Atm^{+/+}$; $Aptx^{+/+}$ mice (P400: 4.5±0.3 [n=9] vs. 4.5±0.2 [n=7] PNs/4000 µm², Welch's t-test, p=0.9; *Figure 5C and D* and *Figure 5—figure supplement 1*). Moreover, we found no evidence from immunohistolological experiments for increased levels of programmed cell death or microglial activation in the $Atm^{R35X/R35X}$; $Aptx^{-/-}$ mice (*Figure 5—figure supplement 2*). At the anatomical level, we found pathological changes in PN morphology, as somatic size was reduced (15.3±0.3 [n=5] vs. 17.8±0.1 [n=5] µm, Welch's t-test, p=0.0004; *Figure 5E*) and the primary and secondary dendrites were abnormally large in caliber in the $Atm^{R35X/R35X}$; $Aptx^{-/-}$ mice (3.1±0.2 [n=6] vs. 2.8±0.1 [n=6] µm, Welch's t-test, p=0.003; *Figure 5F and -Figure 5—figure supplement 2*). Overall, we found a good correlation between the abnormal structural and electrophysiological properties of the cerebellum and the progression of motor and deficits.

## Differential disruption of thymocyte development in ATM-deficient versus APTX-deficient mice

Chronic sinopulmonary infections associated with immunodeficiency are one of the leading causes of death in A-T patients (*Morrell et al., 1986*; *Bhatt and Bush, 2014*). Immunodeficiency is linked to deficits in the generation of B- and T-lymphocytes that have been linked to defects in the antigen receptor gene rearrangement processes during the generation of these cells in the bone marrow and thymus, respectively (*Staples et al., 2008*). The resulting defects in mature lymphocyte numbers include decreases in CD4⁺ helper T-cells and killer CD8⁺ T-cells (*Schubert et al., 2002*). We therefore examined the percentages of T-cells in peripheral blood and of different subpopulations in the thymus of $Atm^{R35X/R35X}$; $Aptx^{-/-}$ mice using T-cell antigen receptor (TCR) and CD4/CD8 co-receptor expression.

In the peripheral blood, we observed a significant reduction in the total fraction of CD3⁺ T-cells in mice with reduced or absent ATM expression compared to wild-type mice (*Figure 6*). This reduction was further compounded by the concomitant deficiency of APTX. ATM and APTX deficiencies reduced T-cells in peripheral blood by over 65% compared to wild-type controls. The effect of APTX deficiency was additive to that of ATM deficiency, suggesting a different mechanism of action for each of these two proteins on T-cell generation. The reduction in the percentage of T-cells in peripheral blood was mostly associated with reduction in the CD4⁺ helper T-cell population (*Figure 6B*). Of interest, the proportion of CD8⁺ T-cells was increased only in $Atm^{R35X/R35X}$; $Aptx^{-/-}$ mice (*Figure 6B*). Again, we observed a differential effect of ATM and APTX deficiencies as seen for the effects of these mutations on the total T-cell fraction.

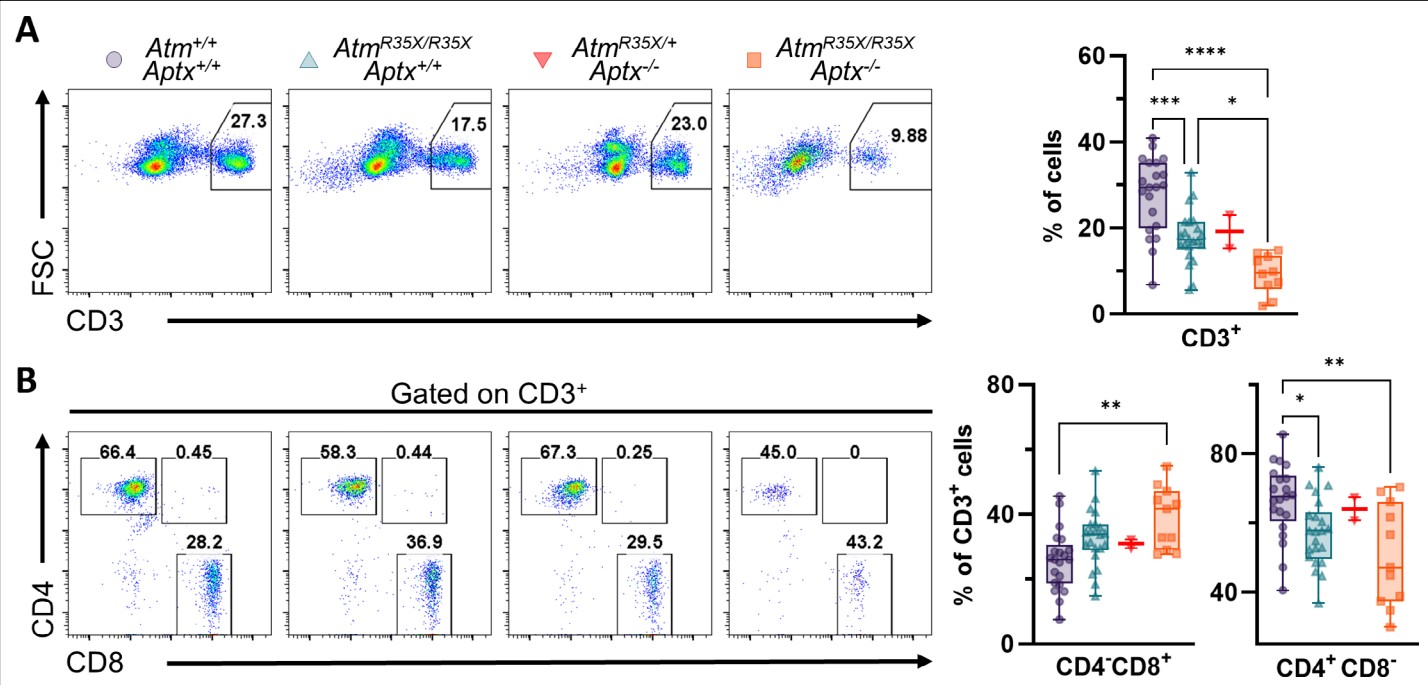

**Figure 6.** T-cell deficits are found in the blood of $Atm^{R35X/R35X}$; $Aptx^{-/-}$ mice. (**A**) Representative flow cytometric profiles of T-cell glycoprotein marker CD3 and summary plots indicate ATM- and/or APTX-deficient mice have decreased proportions of CD3+ T-cells in the blood. (**B**) Representative flow cytometric profiles of T-cell glycoprotein markers CD4 and CD8 gated on CD3+ cells and summary plots for CD8 and CD4 single positive cell proportions. ATM-deficient mice had reduced CD4+ proportions compared to mice with at least one copy of the *Atm* gene. Statistical significances were assessed via one-way ANOVA followed by Tukey's pairwise multiple comparisons test. Number of animals denoted at bottom of bar. Symbol/color key: $Atm^{+/+}$; $Aptx^{+/+}$ (purple circle), $Atm^{R35X/R35X}$; $Aptx^{+/+}$ (green triangle), $Atm^{R35X/+}$; $Aptx^{-/-}$ (red inverted triangle), $Atm^{R35X/R35X}$; $Aptx^{-/-}$ (orange square). ATM, Ataxia Telangiectasia mutated.

The online version of this article includes the following figure supplement(s) for figure 6:

**Source data 1.** Table of numerical FACs data for blood samples.

Given the reduction in T-cell populations in the blood, we next assessed T-cell development in the thymus. In this organ, bone marrow-derived T-cell progenitors undergo TCR gene rearrangement followed by positive selection for MHC restriction and negative selection of autoreactive clones. The phases of thymocyte development can be followed by monitoring the expression of CD4 and CD8 expression in thymocytes. The progression of this developmental program goes from double negative (CD4−CD8−) thymocytes to double positive (CD4+CD8+) thymocytes and then to single positive (CD4+ or CD8+) thymocytes. In addition, within the double negative stage, four different subpopulations can be identified, based on the expression of CD25 and CD44, known as DN1 (CD44+CD25−), DN2 (CD44+CD25+), DN3 (CD44−CD25+), and DN4 (CD44−CD25−) (*Germain, 2002*).

Gene rearrangement during thymocyte development occurs twice—once at the double negative thymocyte stage in the CD25+CD44− stage (*Krangel, 2009*) and then again in double positive thymocyte stage before progressing into separate CD4+ and CD8+ single positive populations (*Livák et al., 1999*). ATM deficiency has been linked to defects in both bouts of rearrangement in mice (*Vacchio et al., 2007*, *Hathcock et al., 2013*). Therefore, we compared the proportion of cells in the thymus expressing these different developmental cell surface markers in our ATM-deficient and control mice (*Figure 7*). $Atm^{R35X/R35X}$; $Aptx^{-/-}$ and $Atm^{R35X/+}$; $Aptx^{-/-}$ but not $Atm^{R35X/R35X}$; $Aptx^{+/+}$ mice had significantly elevated proportions of CD44+CD25−, CD44+CD25+, and CD44−CD25+ cells compared to wildtype (*Figure 7A*). These increased proportions appear to be due in part to an impediment of CD44−CD25+ cells maturing into CD44−CD25− double negative cells, as the fraction of CD44−CD25− cells from $Atm^{R35X/R35X}$; $Aptx^{-/-}$ and $Atm^{R35X/+}$; $Aptx^{-/-}$ mice is significantly lower than wildtype (*Figure 7A*). Of interest, APTX deficiency by itself had the greatest effect on the loss of DN4 cells, suggesting that APTX deficiency, rather than ATM deficiency, is responsible for this effect. To our knowledge,

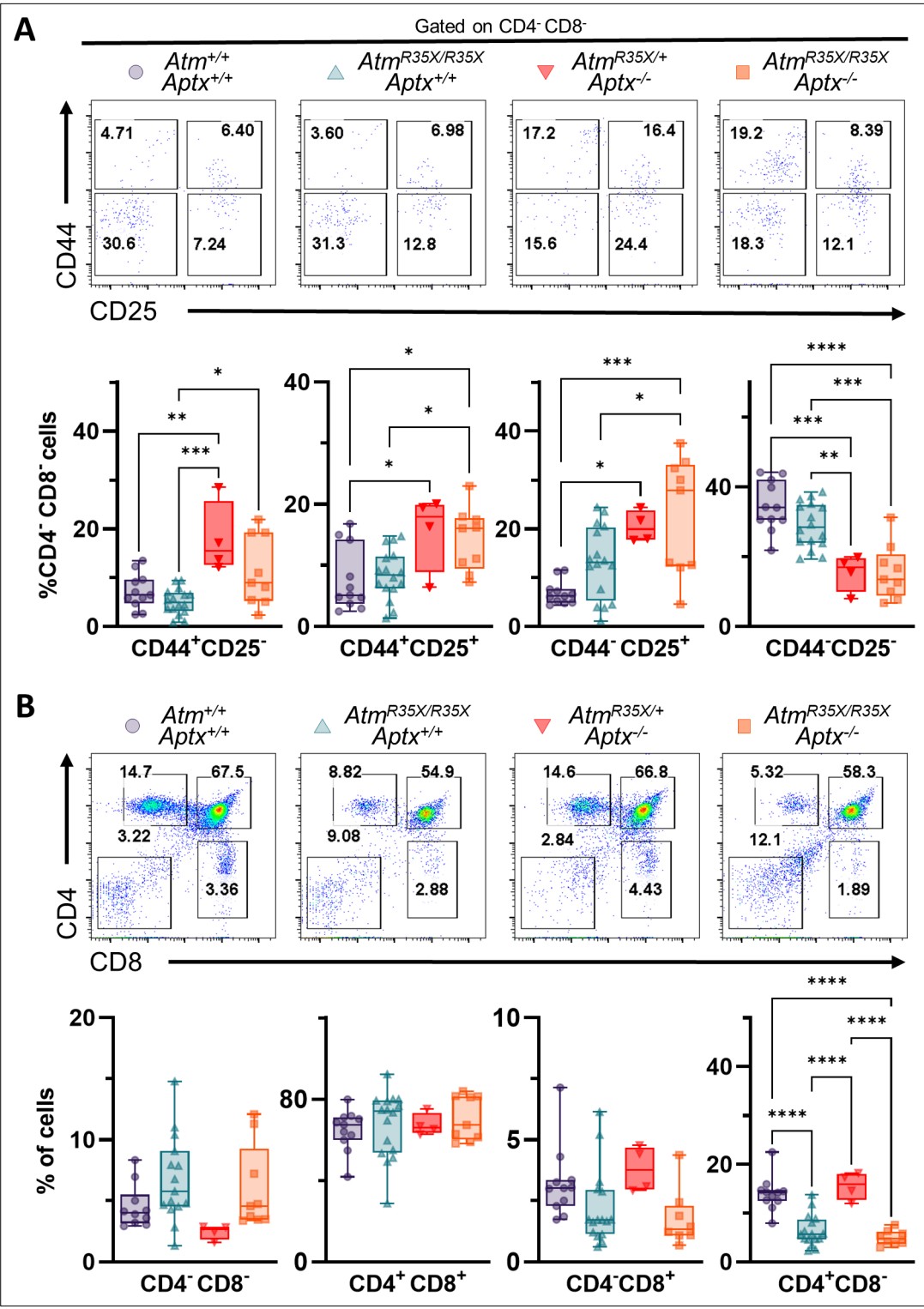

**Figure 7.** ATM and APTX deficiency confer deficits in T-cell expression, but at different developmental stages. (**A**) Representative flow cytometric profiles of T-cell glycoprotein markers CD44 and CD25 gated on CD4⁻CD8⁻ double negative (DN) cells. Summary plots show proportions of thymocytes at DN stages 1–4 (left to right). APTX deficient mice display increased proportions for DN1–3 and decreased proportion at DN4 consistent with a deficit in ontogeny from DN3 to DN4. (**B**) Representative flow cytometric profiles of T-cell glycoprotein markers CD4 and CD8 gated. ATM-deficient mice display decreased proportions for CD4 and CD8 single positive cells consistent with a deficit in ontogeny from CD4⁺CD8⁺ double positive to CD4⁺ and CD8⁺ single positive fates. Statistical

*Figure 7 continued on next page*

*Figure 7 continued*

significances were assessed via one-way ANOVA followed by Tukey's pairwise multiple comparisons test. Number of animals denoted at bottom of bars. Symbol/color key: $Atm^{+/+}$; $Aptx^{+/+}$ (purple circle), $Atm^{R35X/R35X}$; $Aptx^{+/+}$ (green triangle), $Atm^{R35X/+}$; $Aptx^{-/-}$ (red inverted triangle), $Atm^{R35X/R35X}$; $Aptx^{-/-}$ (orange square).

The online version of this article includes the following figure supplement(s) for figure 7:

**Source data 1.** Table of numerical FACs data for thymus samples.

this finding implicates for the first time APTX in gene rearrangement during the process of TCRβ recombination.

Next, we looked at the proportions of CD4$^+$CD8$^+$ thymocytes compared to CD4$^+$CD8$^-$ and CD4$^-$CD8$^+$ single positive thymocytes in these four different strains. In agreement with our results in the blood and prior studies, we found that ATM-deficient mice but not control mice displayed decreased expression of CD4$^+$CD8$^-$ and CD4$^-$CD8$^+$ single positive thymocytes (*Figure 7B*). These results support the role of ATM in TCR α/δgene rearrangement during thymocyte development (*Bredemeyer et al., 2006*), a role that is independent of the role played by APTX in early thymocyte maturation.

## Read-through molecules overcome PTC to restore ATM expression

Our primary rationale for inserting a clinically relevant nonsense mutation in the *Atm* gene was to generate a mouse amenable to critical pre-clinical testing of a novel set of SMRT compounds. We previously demonstrated SMRT compounds recover production of ATM protein in A-T patient-derived

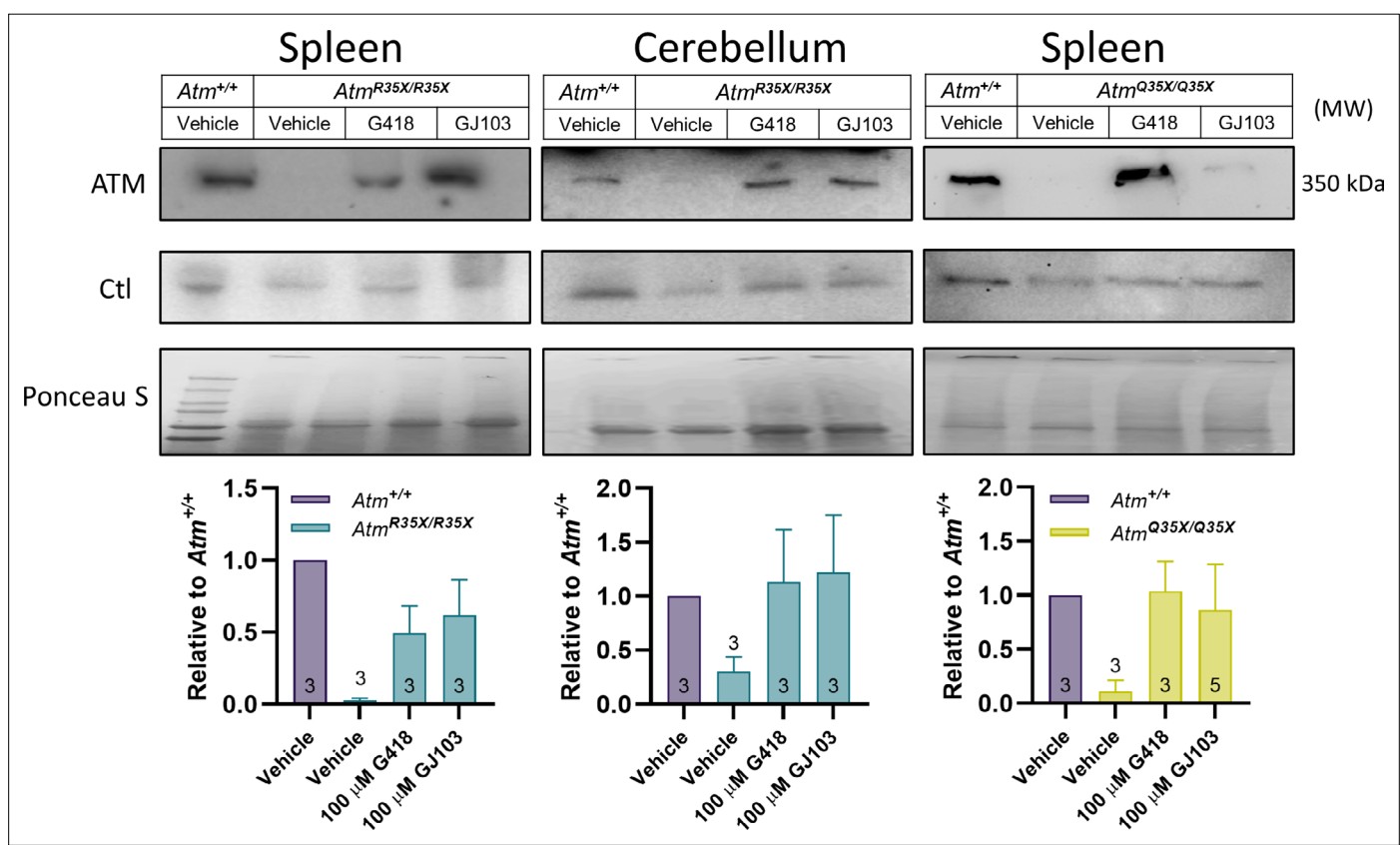

**Figure 8.** ATM protein expression is restored after read-through compound exposure in explant tissues from $Atm^{R35X/R35X}$ and $Atm^{Q35X/Q35X}$ mice. Spleen and cerebellar explant tissue from $Atm^{R35X/R35X}$ and $Atm^{Q35X/Q35X}$ mice were treated with vehicle, the read-through compounds G418 (100 µM) or GJ103 (100 µM) for 72 hr. ATM immunoblots show recovery of ATM (MW 350 kDa) production in both the spleen and cerebellum. Equal loading was assessed via housekeeping genes (Actin or GAPDH) and ponceau staining. ATM, Ataxia Telangiectasia mutated.

The online version of this article includes the following figure supplement(s) for figure 8:

**Source data 1.** Original blots and numerical data.

lymphoblastoid cell lines by overcoming PTC producing nonsense mutations (*Du et al., 2013*). To demonstrate suitability of this new A-T animal model for SMRT compound testing we chose to directly examine their ability to restore ATM expression using an explant approach that circumvents challenges related to in vivo delivery (e.g., bioavailability, route of delivery, etc.). ATM expression was measured in samples from the spleen, where ATM is normally expressed at high levels, and the cerebellum, a key target tissue for the disorder. We exposed these explant tissues, harvested from homozygous *Atm^R35X* and *Atm^Q35X* mice with either a candidate SMRT compound (GJ103), or an aminoglycoside previously known to have read-through properties (G418), and then measured ATM expression by immunoblot to assess restoration. In both types of ATM-deficient mice, ATM expression was consistently restored in the spleen and cerebellum by both G418 and GJ103 (*Figure 8*). These results demonstrate that our SMRT compounds can enable read-through of at least two of the three possible nonsense mutations causing PTCs and provide the rationale for in vivo efficacy testing in follow-on studies.

## Discussion

By increasing genotoxic stress through the addition of a secondary hit to the DDR pathway, we generated a novel mouse model that displays the most comprehensive set of A-T symptoms of any model to date. This includes a severe and progressive ataxia associated with cerebellar atrophy and perturbations of PN properties along with a high incidence of cancer and defects in immune cell development. Taken together, these comorbidities encompass the three leading causes of premature death in A-T—each contributing to roughly a third of deaths. Of these, the incapacitating effect of ataxia is the most penetrant and is reported by patients and caregivers as having the greatest impact on their quality of life. For this reason, the presence of ataxia and cerebellar atrophy in this new mouse model is of great significance, as it provides for the very first time a resource to not only elucidate the mechanisms of neurological dysfunction, but also a critically needed in vivo model to test severely needed A-T therapeutics, such as the read-through compounds we describe here.

We found several similarities between the overall progression of ataxia in the *Atm^R35X/R35X*; *Aptx^−/−* mice and A-T patients. In clinical A-T, motor deficits are observable by roughly 2 years of age, when parents and doctors detect a lowered ability to transition from toddling to a smooth, reflexively coordinated gate—unfortunately, little is known about motor defects at earlier stages due to the diseases low prevalence and current lack of early diagnostic testing (*Rothblum-Oviatt et al., 2016*). Patients usually learn to walk without assistance and neurological symptoms tend to remain stable through the first 4–5 years of life (*Rothblum-Oviatt et al., 2016*). We found a similar early progression of motor deficits in *Atm^R35X/R35X*; *Aptx^−/−* mice, detecting early mild motor deficits at P8 (righting reflex deficit), followed by a period of relative stability, prior to the onset of a progressive and severe ataxia developing after P210 that included changes in gait, startle reflex, tremor, and locomotor activity. Several important questions arise out of these findings, including whether ATM and/or APTX have a neurodevelopmental role in the cerebellum. Future studies focused on the early phase of the disorder will be critical in understanding if the cerebellum develops normally prior to dysfunction or whether developmental defects are an initial cause. We also found, similar to A-T patients, that the severity of the late-developing ataxia was variable, with some mice ambulating with a clumsy, high-stepping rear gate (*Video 3*) and others moving almost entirely via contortion of the rear trunk (*Figure 1E* and *Video 4*; *Rothblum-Oviatt et al., 2016*; *Levy and Lang, 2018*; *Boder and Sedgwick, 1958*). Overall, we found that *Atm^R35X/R35X*; *Aptx^−/−* mice developed a visually profound and measurable progressive loss in motor coordination similar to that observed in A-T patients, which was rescued by expression of at least one copy of the *Atm* or *Aptx* gene.

The loss of motor coordination in A-T has been attributed to cerebellar degeneration due to its relatively selective neuropathology across the brain and its causal role in several different forms of ataxia (*Hoche et al., 2012*). Consistent with A-T patient neuroimaging studies (*Wallis et al., 2007*; *Sahama et al., 2015*; *Sahama et al., 2014*; *Dineen et al., 2020*; *Tavani et al., 2003*; *Quarantelli et al., 2013*), we find that cerebellar size in *Atm^R35X/R35X*; *Aptx^−/−* mice is initially normal, but progressively atrophies concurrently with changes in neurological function. While loss of cerebellar tissue has been considered a main cause of ataxia in humans, it is unclear from clinical data if ataxia severity is a good predictor of the extent of cerebellar degeneration found postmortem (*Aguilar et al., 1968*; *Crawford et al., 2006*; *Dineen et al., 2020*). In the *Atm^R35X/R35X*; *Aptx^−/−* mice, we find clear atrophy associated with thinning of the PN dendrite layer that precedes the late, severe behavioral deficits.

Our histological observations in the $Atm^{R35X/R35X}$; $Aptx^{-/-}$ mice suggest that changes in cerebellar function itself, rather than profound loss of cerebellar cells, are sufficient to cause the ataxic phenotype, consistent with the observation of behavioral defects prior to significant PN loss in several SCAs (*Shakkottai et al., 2011*; *Lorenzetti et al., 2000*; *Clark et al., 1997*; *Jayabal et al., 2016*).

The reason why ATM and APTX deficiency is required to generate ataxia in mice, when loss of either is sufficient to cause ataxia in humans, remains unclear. One possibility is that the rodent brain may more flexibly utilize compensatory pathways or redundant proteins while responding to the ~15k DNA lesions that impact cells each day (*Lindahl and Barnes, 2000*). Several forms of DNA repair exist to potentially meet this challenge, including base excision repair (BER), nucleotide excision repair (NER), as well as homologous and non-homologous end joining (HEJ and NHEJ, respectively), all of which ATM and APTX have been implicated in (*Chou et al., 2015*; *Caglayan et al., 2017*; *Wakasugi et al., 2014*; *Tumbale et al., 2018*; *Chatterjee and Walker, 2017*). Alternatively, it may be the case that deficiency in ATM or APTX alone does not adequately impact cell health during the mouse's comparatively short lifespan, and thus eliminating both proteins is necessary to achieve sufficient accumulation of DNA damage to manifest over this time period. This possibility is strengthened by the fact that ATM and APTX have distinct biochemical properties and functional roles in the DDR, and therefore deficiency in both would be predicted to cause a broader, additive hit to genome stability than either alone (i.e., increased genotoxic stress).

Our finding that two genome stability pathway proteins are required to induce neurological defects in mice strongly suggests that it is the loss of ATM's role in DNA repair, rather than potential functions in oxidative stress signaling, mitophagy, or mitochondrial function that cause the cerebellar defects (*Shiloh, 2020*). Alternatives, however, cannot be completely ruled out, as APTX, like ATM, has been observed within the mitochondria of brain cells, where it is thought to support the processing of mitochondrial DNA (*Meagher and Lightowlers, 2014*; *Sykora et al., 2011*). This new mouse model provides a new tool to explore these possibilities and mechanistically define how loss of ATM and APTX ultimately causes cerebellar dysfunction.

The biophysical perturbations observed in PNs recorded from the $Atm^{R35X/R35X}$; $Aptx^{-/-}$ mice are similarly found in several other mouse models of ataxia. This includes changes we observed in PN input resistance, membrane capacitance, and AP threshold and width, which have also been described in mouse models of SCA like 1, 3, and 7 (*Stoyas et al., 2020*; *Shakkottai et al., 2011*; *Dell'Orco et al., 2015*). Moreover, the progressive reduction in PN AP firing frequency we report, which positively correlates with the development of ataxia in the $Atm^{R35X/R35X}$; $Aptx^{-/-}$ mice, is reported in a large number of ataxic mouse models, including SCAs 1, 2, 3, 5, 6, and 13 as well as a few episodic forms (see review *Cook et al., 2021*).

Given the significant overlap in PN perturbations observed across many different ataxias caused by distinct cellular defects, restoring PN AP firing frequencies has been considered as a broad-based therapeutic approach. However, it remains unclear whether reduced PN firing is a causal factor of ataxia. Moreover, experimental evidence suggests changes in PN activity may in fact be a generalized response to maintain homeostasis during ongoing disease-related impairment of PN physiology (*Dell'Orco et al., 2015*). Thus, continued efforts across all cerebellar ataxias are needed to link the genetic, molecular, and cellular disruptions caused by disease to the specific changes in cerebellar neural signaling that ultimately generates the ataxia. Of significant importance in this effort will be determining whether disease-causing cerebellar defects commonly or differentially cause ataxia through a loss of cerebellar function (e.g., loss of coordinating signals during movement), or from a dominant negative effect (e.g., disrupting downstream neural circuits with abnormal neural output patterns). Ultimately, while a common therapeutic strategy to address cerebellar ataxias would have the greatest impact, a directed approach that addresses the distinct genetic and molecular causes of cellular dysfunction may ultimately be necessary to successfully develop an efficacious therapeutic.

The mechanistic link between deficiency in DNA stability proteins like ATM and APTX and PN dysfunction is far from clear. Our results suggest the effect of ATM and APTX loss on PNs is intrinsic, as we do not find changes in the presynaptic properties of granule cells or evidence of their cellular loss (no change in GCL thickness). Moreover, while we observed differences in short term plasticity of inferior olivary inputs in ATM- and APTX-deficient PNs and wildtype, these results likely point to a disruption in $Ca^{2+}$ homeostasis potentially via reductions in Inositol 1,4,5-triphosphate receptor 1 (*Itpr1*) expression, similar to those observed in SCAs 1, 2, and 3 mouse models as well as ATM-deficient

mice (*Kim et al., 2020*; *Chen et al., 2008*; *Demirci et al., 2009*; *Shakkottai et al., 2011*). While this provides a promising avenue for future examination and comparison, it is as of yet unclear, even for the SCAs, whether changes in $Ca^{2+}$ homeostasis, is the causal factor or just another symptom or even compensatory response of diseased or disturbed PNs (*Dell'Orco et al., 2015*).

In the immune system, ATM is implicated in the repair of DNA breaks that naturally occur during gene rearrangement of antigen receptor genes in B- and T-cell precursors, a phenomenon critical for antigen receptor (Ig and TCR) diversity of these cells. Our finding that T-cell proportions in the blood are significantly reduced is consistent with prior studies in humans and A-T knockout mice (*Schubert et al., 2002*; *Hathcock et al., 2013*; *Chao et al., 2000*; *Barlow et al., 1996*). This reduction of T-cells in the periphery likely correlates with a defect in both cellular and humoral immunity. Importantly, we found that expression of one copy of the ATM gene is enough to restore $CD4^+$ deficits in the blood indicating that therapies able to restore partial ATM expression would have therapeutic efficacy. Although we have not assessed B-cell development in this paper, it is likely that similar conclusions would apply to that process given their mechanistic similarities (*Marshall et al., 2018*).

As expected, the reduction of T-cells in peripheral blood is correlated with defective thymocyte development. In the thymus, we found two main defects. One, induced primarily by APTX deficiency, manifests as a defect in the DN3 to DN4 transition coinciding with early rearrangement of TCR **β** locus. The other defect, primarily caused by ATM deficiency, correlates with decreased progression of double positive $CD4^+CD8^+$ to single positive cells, primarily $CD4^+$ thymocytes. While the APTX finding was surprising, as its deficiency (AOA 1) is not associated with immune deficits, APTX is known to interact with TCR **β** gene rearrangement proteins, including XRCC4 (*Clements et al., 2004*). Future studies aimed at defining APTX's role in end-joining mechanisms during TCR gene rearrangement will be important, and the possibility that alternative end-joining mechanisms, like the use of microhomologies account for the lack of an immune deficit in its absence needs further investigation (*Bogue et al., 1997*).

The survivability of $Atm^{R35X/R35X}$; $Aptx^{-/-}$ mice is considerably longer than prior A-T mouse models. In comparison, the first A-T KO mouse model reported by *Barlow et al., 1996* died from thymomas usually within 2–4 months after birth. The decreased cancer survivability in this and many other knockout A-T mouse models is likely genetic, as the background strain harboring the mutation has been shown to have significant effects on cancer prevalence and survivability, with A/J and C57BL/6 backgrounds having significantly increased survivability over the BALBC and 129S strains (*Genik et al., 2014*). The fact that our ATM-deficient mice were created on a C57BL/6 background likely underlies their comparatively long lifespan. Given that the $Atm^{R35X/R35X}$; $Aptx^{+/+}$ mice do not develop ataxia, it is unlikely that the early death in A-T KO mice prevents observation of an ataxic phenotype that would otherwise develop in these mice. However, it is unknown whether the C57BL/6 background confers a resilience to developing ataxia, as it does for cancer. Defining the genetic or possibly epigenetic factors that influence the severity of the disease could provide avenues for future therapeutic development.

Given the global nature of the ATM and APTX null mutation in our mouse model, we cannot entirely rule out that extra-cerebellar defects may also contribute to the severe ataxic phenotype, and thus future examination outside the cerebellum in the forebrain, brainstem, spinal cord, and even muscle will need to be conducted. Within the cerebellum, while we found some anatomical differences in the PN firing properties within different regions of the cerebellum, we did not detect regional differences in ML width or PN density. However, there are challenges in using regional anatomy as a grouping factor in the cerebellum, as the physical folds of the tissue do not necessarily correlate with the boundaries of functional, molecular expression, or physiological property domains that have been described (*Apps and Hawkes, 2009*; *Tsutsumi et al., 2015*; *Gao et al., 2012*; *Zhou et al., 2014*). Experiments focused on examining the extent of cerebellar defects within these domains will be important in future studies and compared to the anecdotal reports of anatomical differences in A-T patients (*Verhagen et al., 2012*; *De León et al., 1976*; *Amromin et al., 1979*; *Monaco et al., 1988*; *Terplan and Krauss, 1969*; *Strich, 1966*; *Solitare, 1968*; *Solitare and Lopez, 1967*; *Aguilar et al., 1968*; *Paula-Barbosa et al., 1983*).

While we detect two potential stages in the progression of ataxia in the $Atm^{R35X/R35X}$; $Aptx^{-/-}$ mice, the later stage of severe ataxia develops in adulthood in mice, as compared to the childhood onset in humans. This may limit its use in some neurodevelopmental studies. Also, the interpretation of future

experiments must carefully factor in the fact that this new model expresses null mutations in two genome stability genes at the same time, a situation that has not been detected in human patients with either A-T or AOA1.

Finally, pinpointing where, when, and how ATM deficiency causes cerebellar pathology and ataxia has been a challenge, as prior ATM-deficient mice generally lack the characteristic features needed to causally link cellular and molecular deficits to the ataxic phenotype. Multiple promising avenues of investigation have been defined, including those focused at the neuronal level, where ATM is implicated in oxidative stress signaling (*Chen et al., 2003*) and synaptic function (*Li et al., 2009*; *Vail et al., 2016*), as well as glial function, where recent evidence suggests glial pathology may be a leading factor in cerebellar pathology (*Kaminsky et al., 2016*; *Campbell et al., 2016*; *Petersen et al., 2012*; *Weyemi et al., 2015*). This novel animal model provides a new tool to test mechanistic hypotheses regarding how ATM deficiency causes cerebellar pathology and ataxia. Additionally, this model may serve most importantly as a critical preclinical tool for testing previously proposed therapeutic candidates (*Browne et al., 2004*; *Chen et al., 2003*) including our own SMRT compounds (*Du et al., 2013*). It cannot be overstated how severely limiting the lack of a preclinical model is for therapeutic development, especially for rare disorders like A-T and AOA1.

# Materials and methods

## Key resources table

| Reagent type (species) or resource | Designation | Source or reference | Identifiers | Additional information |
|---|---|---|---|---|
| Strain, strain background (*Mus musculus*) | $Atm^{R35X}$, $Atm^{Tm1.1(103CAG)TGAMfgc}$ | This paper | 103C>T mutation, human exon replacement | Generated by Hicks laboratory. Has been backcrossed into C57b/6 9 times. Contact pmathews@lundquist.org |
| Strain, strain background (*M. musculus*) | $Atm^{Q35X}$, $Atm^{Tm1.1(103C)TMfgc}$ | This paper | 103C>T mutation, targeted premature termination signal in the mouse codon | Generated by Hicks laboratory. Has been backcrossed into C57b/6 9 times. Contact pmathews@lundquist.org |
| Strain, strain background (*M. musculus*) | $Aptx^{-/-}$ | *Ahel et al., 2006* | MGI Cat# 3687171, RRID:MGI:3687171 | Contact peter.mckinnon@stjude.org |
| Gene (*M. musculus*) | Atm | MGI | MGI:107202; C030026E19Rik; ENSMUSG00000034218 | |
| Gene (*Homo Sapien*) | ATM | OMIM | OMIM: 607585 MGI: 107202 HomoloGene: 30952; ENSG00000149311 | |
| Sequence-based reagent | Atm gene | Transnetyx | PCR primers | F-5'-CCTTTGAGGCATAAGTTGCA ACTTG-3' |
| Sequence-based reagent | Atm gene | Transnetyx | PCR primers | R-5'-GTACAGTGTATCAGGTTAGG CATGC-3' |
| Chemical compound/ drugs | GJ103 salt Formula: $C_{16}H_{14}N_4O_3S$ | TargetMol | T3448; CAS No.: 1459687-96-7 | 100 µM in media |
| Antibody | Anti-mouse CD68 (Rat monoclonal) | Bio-Rad | Cat# MCA1957, RRID: AB_322219 | IF (1:400) |
| Antibody | ATM (D2E2) (Rabbit-monoclonal) | Cell Signaling Technology | Cell Signaling Technology Cat# 2873, RRID:AB_2062659 | WB (1:500) WB (1:1000) |
| Antibody | GAPDH (14C10) (Rabbit-monoclonal) | Cell Signaling Technology | Cell Signaling Technology Cat# 2118, RRID:AB_561053 | WB (1:4000) |
| Antibody | β-Actin (D6A8) (Rabbit-monoclonal) | Cell Signaling Technology | Cell Signaling Technology Cat# 8457, RRID:AB_10950489 | WB (1:5000) |
| Antibody | Anti-Rabbit IgG, HRP-linked (Goat-monoclonal-polyclonal) | Cell Signaling Technology | Cell Signaling Technology Cat# 7074, RRID:AB_2099233 | WB (1:5000) |
| Antibody | Anti-Calbindin (D-28k) (Rabbit, polyclonal) | Swant Inc. | Swant Cat# CB 38, RRID:AB_10000340 | IF (1:1000) |

*Continued on next page*

*Continued*

| Reagent type (species) or resource | Designation | Source or reference | Identifiers | Additional information |
|---|---|---|---|---|
| Antibody | Anti-Mouse Alexa Fluor 488 (Goat polyclonal) | Thermo Fisher Scientific; Invitrogen | Cat# A11001, RRID: AB_2534069 | IF (1:500) |
| Antibody | Anti-mouse Cleaved Caspase-3, Asp-175 (rabbit) | Cell Signaling Technology | Cat# 9961, RRID: AB_2341188 | IF (1:200) |
| Antibody | Anti-Rat Alexa Fluor 555 (Goat polyclonal) | Thermo Fisher Scientific; Invitrogen | Cat# A21244, RRID: AB_2535812 | IF (1:1000) |
| Antibody | Anti-Rabbit Alexa Fluor 647 (Goat polyclonal) | Thermo Fisher Scientific; Invitrogen | Cat# A-21434, RRID: AB_141733 | IF (1:500) |
| Antibody | Anti-Calbindin D-28k (mouse-monoclonal) | Swant Inc. | Cat# CB300 | IF (1:500) |
| Antibody | Anti-Rabbit Alexa Fluor 488 (Goat-polyclonal) | Thermo Fisher Scientific; Invitrogen | Thermo Fisher Scientific Cat# A-11034, RRID:AB_2576217 | IF (1:1000) |
| Antibody | CD4 (GK1.5) (Rat-monoclonal) | Thermo Fisher Scientific; Invitrogen | Thermo Fisher Scientific Cat# 50-0041-82, RRID:AB_10609337 | FACS (5 µl per test) |
| Antibody | CD8 (53-6.7) (Rat-monoclonal) | Thermo Fisher Scientific; Invitrogen | Thermo Fisher Scientific Cat# 53-0081-82, RRID:AB_469897 | FACS (5 µl per test) |
| Antibody | CD3 (145- 2C11) (Hamster-monoclonal) | Thermo Fisher Scientific; Invitrogen | Thermo Fisher Scientific Cat# 12-0031-83, RRID:AB_465497 | FACS (5 µl per test) |
| Antibody | CD44 (IM7) (Rat-monoclonal) | Thermo Fisher Scientific; Invitrogen | Thermo Fisher Scientific Cat# 25-0441-82, RRID:AB_469623 | FACS (5 µl per test) |
| Antibody | CD25 (PC61.5) (Rat-monoclonal) | Thermo Fisher Scientific; Invitrogen | Thermo Fisher Scientific Cat# 47-0251-82, RRID:AB_1272179 | FACS (5 µl per test) |
| Other | Eosin Y (Certified Biological Stain) | Thermo Fisher Scientific (Fisher Chemical) | Cat# E511-100 | |
| Other | Hematoxylin Stain Solution, Modified Harris Formulation, Mercury Free Nuclear Stain | RICCA Chemical Company | Cat# 3530-16 | |
| Other | Permount Mounting Medium | Thermo Fisher Scientific (Fisher Chemical) | Cat# SP15-100 | |
| Other | Fluoromount-G with DAPI | Southern Biotech | Cat# 0100-20, RRID: SCR_021261 | |
| Commercial assay or kit | BCA Protein Assay Kit | Thermo Fisher Scientific; Pierce | Cat# 23225 | Protein assay |
| Commercial assay or kit | SuperSignal West Pico Chemiluminescent Substrate | Thermo Fisher Scientific; Pierce | Cat# 34580 | Chemiluminescent substrate |
| Commercial assay or kit | Radiance plus | Azure Biosystems | Cat# AC2103 | Chemiluminescent substrate |
| Software, algorithm | FlowJo | https://www.flowjo.com/solutions/flowjo | RRID:SCR_008520 | |
| Software, algorithm | ImageJ software | ImageJ (http://imagej.nih.gov/ij/) | RRID:SCR_003070 | Version 1.53 |

*Continued on next page*

*Continued*

| Reagent type (species) or resource | Designation | Source or reference | Identifiers | Additional information |
|---|---|---|---|---|
| Software, algorithm | IgorPro | http://www.wavemetrics.com/products/igorpro/igorpro.htm | RRID:SCR_000325 | Version 7; Tarotools procedures |
| Software, algorithm | Neuroexpress | https://www.researchgate.net/project/NeuroExpress-Analysis-software-for-whole-cell-electrophysiological-data | https://www.researchgate.net/project/NeuroExpress-Analysis-software-for-whole-cell-electrophysiological-data | Version 21.1.13; used for sEPSC analyses |
| Software, algorithm | GraphPad, Prism | GraphPad Prism (https://graphpad.com) | RRID:SCR_015807 | Versions 8 and 9 |
| Software, algorithm | MBF, Stereo investigator | https://www.mbfbioscience.com/stereology | RRID:SCR_017667 | Version 2021 |
| Software, algorithm | Microsoft Excel | https://www.microsoft.com/en-us/microsoft-365/excel | RRID:SCR_016137 | Version 365 |
| Software, algorithm | Catwalk XT | https://www.noldus.com/catwalk-xt | RRID: SCR_021262 | |

## Mice

All mice were group housed and kept under a 12 hr day/night cycle with food and water available ad libitum. Animals were housed within the general mouse house population, and not in specialized pathogen-free rooms. Older animals were made available wetted food or food gel packs on the ground of the cages as ataxia developed. *Atm^{R35X}* and *Atm^{Q35X}* mice were created and provided by Dr. Hicks and colleagues at the University of Manitoba.

These mice were created to contain the c.103C>T mutation found in a large population of North African AT patients using recombineering Gateway technology and site-directed mutagenesis. A C>T mutation at this position in the mouse *Atm* gene creates a TAG G stop codon. The same mutation in the human ATM gene produces a TGA G stop codon. In consideration of the use of these models for therapeutic interventions, we chose to create a mouse model for each of the two PTC codons (**Figure 1A**).

A modified Gateway R3-R4-destination vector was used to pull out the desired region of the mouse *Atm* gene from a Bacterial Artificial Chromosome (BAC) and subsequently mutated to create either a TAG G stop codon at codon 35 (M00001, position 103 [C>T]) or a TGA G stop codon (M00002, position 103 (CAG>TGA), replicating the human AT PTC. The genomic alleles were then cloned into a modified version of the NorCOMM mammalian targeting vector using a three-way Gateway Reaction (**Bradley et al., 2012**). The resulting targeting vectors were electroporated into C2 ES cells (C57Bl/ 6N, derived in A. Nagy lab, Toronto, Canada) and successfully targeted clones were identified by selection with G418 (**Gertsenstein et al., 2010**). Integration of the mutated targeting cassette into the *Atm* gene locus was confirmed by Southern blot, and by sequencing of PCR products to confirm the presence of the *Atm* PTC mutation, error-free targeting into the *Atm* locus, and error-free functional components of the vector (data not shown). Positive ES clones were used for blastocyst injection to obtain the transgenic lines. The transgenic allele contained a floxed human beta actin promoter—delta TK1—Neo cassette in the intron upstream of the region containing the mutated exon. This floxed cassette was subsequently excised by crossing with a Cre driver mouse (B6.C-Tg(CMV-cre)1Cgn/J) to generate *Atm^{R35X/+}* and *Atm^{Q35X/+}* (MGI nomenclature: *Atm^{TM1(103CAG>TGA)MFGC}* and *Atm^{TM1(103C>T)MFGC}*, respectively) mouse lines (**Figure 1A**). Genotyping of the two *Atm* lines was performed by using the following primers at Tm 62°C: *Atm* gene forward (F) primer: 5'-CCTTTGAGGCATAAGTTGCAACTTG-3'; and *Atm* gene reverse (R) primer: 5'-GTACAGTGTATCAGGTTAGGCATGC-3', creating a wild-type allele product of 151 bp or targeted allele product of 241 bp (**Figure 1A and B**).

$Atm^{R35X}$ and $Atm^{Q35X}$ were back-crossed with C57Bl/ 6J mice for nine generations ( 99.2% isogenic) prior to cryopreservation and subsequent rederivation using C57Bl/ 6J surrogate mothers. $Atm^{R35X}$ and $Atm^{Q35X}$ breeders were obtained from F1 sibling $Atm^{R35X/+}$ and $Atm^{Q35X/+}$ mice. $Atm^{R35X/R35X}$ and $Atm^{Q35X/Q35X}$ were both found to be fertile. $Aptx$ knockout ($Aptx^{-/-}$) mice were created and provided to Dr. Mathews as embryos from Dr. McKinnon (**Ahel et al., 2006**), and subsequently rederived via C57Bl/ 6J surrogate mothers. $Aptx^{-/-}$ mice are on a C57Bl/6 and 129 mixed background. $Atm^{R35/R35XX}$; $Aptx^{-/-}$ mice of various wildtype, heterozygous, and homozygous combinations were created from $Atm^{R35X/+}$; $Aptx^{+/-}$ generated by crossing $Atm^{R35X/R35X}$ and $Aptx^{-/-}$ mice. One cohort of double mutant and corresponding control mice were used in the longitudinal behavioral study for gait analyses and SHIRPA testing (**Figures 2 and 3**). Multiple additional cohorts of age-matched double mutant and control mice were used for electrophysiological, immunohistological, and Vertical Pole test experiments (**Figures 4 and 7**). Immunological and protein expression experiments were carried out using mice bred from the original $Atm^{R35X}$ and $Atm^{Q35X}$ rederived mice (**Figures 5, 6 and 8**).

Genotyping was performed from ear tissue samples of P8-11 mice. Real-time PCR methods conducted by Transnetyx Inc were used to determine each animals' genotype. Animals were made identifiable via toe tattoos given at the same time as ear biopsy. Unique primers for $Atm^{R35X}$ and $Atm^{Q35X}$ were quantified and used to identify wildtype, heterozygous and homozygous mice (listed above). $Aptx^{-/-}$ and $Aptx^{+/+}$ primers were used to assess their genotypes.

## Animal health

Animals were weighed via a digital scale at P8, P45, P120, P210, and P400. Animal death was recorded as the day found dead, or on the day of euthanization when the animals reached a humane endpoint (animal unable to right itself within 60 s, significant hair matting indicating lack of self-grooming, or excessive distress as noted by the veterinary staff). Animal carcasses were immediately frozen upon death, and postmortem necropsies were carried out in batch. Probable cause of death was determined to the best of our ability in collaboration with the staff veterinarian (Dr. Catalina Guerra) by visual inspection of the internal organs. Some mice were cannibalized or accidentally disposed of by vivarium staff and were therefore labeled 'missing.' Mice with no discernable visual cause of death were labeled 'indeterminable.' Mice that were found with thoracic masses near where the thymus would normally be in young mice were listed as 'thymic cancer.' All other identified probable causes of death (e.g., enlarged livers, urinary blockage) were labeled 'other.'

## Behavior

Before performing any behavioral test, mice were acclimated to the behavioral suite for ~20 min. Mice were tested at varying times of the day, in line with their day cycle. A battery of behavioral tests were performed on naïve double mutant mice of the indicated genotypes at various time points depending on the behavior but in the same cohort of mice. The battery of tests included Catwalk Gait assessment (P45, P120, P210, and P400) and a subset of the SmithKline-Beecham Harwell Imperial-College and Royal-London-Hospital Phenotype Assessment (SHIRPA) tests (P30 and P400). These tests were conducted by the UCLA Behavioral Core. Double mutant and control mice were additionally examined on the Vertical Pole test. All behavioral apparatuses were wiped down with ethanol (70%) between each testing each subject.

### Gait analysis

We used a Noldus Catwalk Gait analysis system designed to semi-automatically measure and analyze the gait of mice during normal ambulation. Briefly, the movement of mice across a glass bottom corridor is video recorded from a ventral position. Paw prints are highlighted in the video due to light illumination across the glass walking platform. Each mouse step within a video is subsequently detected using Catwalk XT (Noldus) in a semi-automated fashion. A run for each mouse consists of three trials of consistent ambulation across the monitored platform. Only consistent trials are accepted, and mice may take up to 10 attempts to complete three compliant trials in either direction across the corridor. Compliant trials were defined as those with movement across the platform under 5-s long and with no more than 60% speed variation. Once placed onto the platform, mice generally ran back and forth without any need for experimenter prompting.

## Vertical pole

Mice were placed at the top of an 80-cm-tall bolt with their nose facing down and hind paws as close to the top as possible. Mice were immediately released, and time started immediately upon placement. Time was stopped when the first forepaw touches the surface below the pole. A mouse's natural predilection is to immediately climb down the pole, and they were given up to 60 s to traverse the pole, otherwise they were helped off the pole. A non-completed trial was automatically given a time of 30 s, as 95% of mice that did not descend within 30 s were still on the pole at the 60 s mark.

## SHIRPA

Behavioral tests were conducted by the University of California, Los Angeles Behavioral Core at P30 and P400. All parameters are scored to provide a quantitative assessment, which enables the comparison of results both over time and between different laboratories. Each mouse was sequentially tested across all behaviors within ~20 min time span before moving onto the next mouse. The experimenter was blinded to animal genotype. The screen was performed as described previously (*Rogers et al., 1997*).

## Behavioral observation

The primary screen provides a behavioral observation profile and assessment of each animal begins by observing undisturbed behavior in a viewing jar (10 cm diameter) for 5 min. In addition to the scored behaviors of body position, spontaneous activity, respiration rate, and tremor, the observer logs any instances of bizarre or stereotyped behavior and convulsions, compulsive licking, self-destructive biting, retropulsion (walking backwards), and indications of spatial disorientation.

## Arena behavior

Thereafter, the mouse was transferred to the arena ($30 \times 50$ cm$^2$) for testing of transfer arousal and observation of normal behavior. The arena was marked into a grid of $10 \times 10$ cm$^2$ squares to measure locomotor activity within a 30-s period. While the mouse was active in the arena, measures of startle response, gait, pelvic elevation, and tail elevation are recorded.

## Supine restraint

The animal was restrained in a supine position to record autonomic behaviors. During this assessment, grip strength, body tone, pinna reflex, corneal reflex, toe pinch, wire maneuver, and heart rate were evaluated.

## Balance and orientation

Finally, several measures of vestibular system function were performed. The righting reflex, contact righting reflex, and negative geotaxis tests were performed. Throughout this procedure vocalization, urination and general fear, irritability, or aggression were recorded.

## Equipment used

1. Clear Plexiglas arena (approximate internal dimensions $55 \times 33 \times 18$ cm$^3$). On the floor of the arena is a Plexiglas sheet marked with 15 squares (11 cm). A rigid horizontal wire (3 mm diameter) is secured across the rear right corner such that the animals cannot touch the sides during the wire maneuver. A grid ($40 \times 20$ cm$^2$) with 12 mm mesh (approximate) is secured across the width of the box for measuring tail suspension and grip strength behavior.
2. A clear Plexiglas cylinder ($15 \times 11$ cm$^2$) was used as a viewing jar.
3. One grid floor ($40 \times 20$ cm$^2$) with 12 mm meshes on which viewing jars stand.
4. Four cylindrical stainless-steel supports (3 cm high × 2.5 cm diameter) to raise grids off the bench.
5. One square (13 cm) stainless steel plate for transfer of animals to the arena.
6. Cut lengths of 3/0 Mersilk held in the forceps for corneal and pinna reflex tests.
7. A plastic dowel rod sharpened to a pencil point to test salivation and biting.
8. A pair of dissecting equipment forceps, curved with fine points (125 mm forceps, Philip Harris Scientific, Cat. no. D46-174), for the toe pinch.
9. A stopwatch.

10. An IHR Click box is used for testing the startle responses. The Click Box generates a brief 20 KHz tone at 90 dB SPL when held 30 cm above the mouse. Contact Prof. K.P. Steel, MRC Institute of Hearing Research, University Park, Nottingham NG7 2RD.
11. A ruler.
12. A 30 cm clear Plexiglas tube with an internal diameter of 2.5 cm for the contact righting reflex.

## Electrophysiology

### Preparation of acute cerebellar slices

Acute parasagittal slices of 300 µm thickness were prepared from the cerebellum of experimental and control littermate mice by following published methods (*Hansen et al., 2013*). In brief, cerebella were quickly removed and immersed in an ice-cold extracellular solution with composition of (mM): 119 NaCl, 26 NaHCO$_3$, 11 glucose, 2.5 KCl, 2.5 CaCl$_2$, 1.3 MgCl$_2$, and 1 NaH$_2$PO$_4$, pH 7.4 when gassed with 5% CO$_2$/ 95% O$_2$. Cerebella were sectioned parasagittally using a vibratome (Leica VT-1000, Leica Biosystems, Nussloch, Germany) and initially incubated at 35°C for ~30 min, and then equilibrated and stored at room temperature until use.

### Extracellular electrophysiology

Extracellular and intracellular recordings were obtained from PNs in slices constantly perfused with carbogen-bubbled extracellular solution and maintained at either 37°C (extracellular) or 32°C (intracellular) ± 1°C (see above). Cells were visualized with DIC optics and a water-immersion 40× objective (NA 0.75) using a Zeiss Examiner microscope. Glass pipettes of ~3 MΩ resistance (Model P-1000, Sutter Instruments, Novato, CA) were filled with extracellular solution and positioned near PN axon hillocks in order to measure AP-associated capacitive current transients in voltage clamp mode with the pipette potential held at 0 mV. For whole-cell patch-clamp recordings, pipettes were filled with an intracellular solution (mM): 140 KMeth (CH$_3$KO$_3$S), 10 NaCl, 2 MgCl$_2$, 0.2 CaCl$_2$, 10 HEPES, 14 Phosphocreatine (tris salt), 1 EGTA, 4 Mg-ATP, 0.4 Na-GTP. 100 µM Picrotoxin (Sigma-Aldrich) was added to block inhibitory GABAegeric synaptic inputs. Data was acquired using a MultiClamp 700B amplifier at 20 or 100 kHz in voltage or current clamp mode, Digidata 1440 with pClamp10 (Molecular Devices, Sunnyvale, CA), and filtered at 2–4 kHz. The series resistance was usually between 10 and 15 MΩ. Series resistance was compensated at 80% for short-term plasticity experiments only.

For extracellular recordings, a total of 20–45 PNs were recorded for each animal across all genotypes, sexes, and age groups. Recordings were distributed across both the medial-lateral and rostrocaudal axis of the cerebellum. Only cells with a 'healthy' look (low contrast of cellular borders) and regular, uninterrupted firing rate were recorded. During analysis, a few cells were found to have gaps in firing of greater than 2 s, and these cells were eliminated from analysis, as this type of firing is associated with being 'unhealthy.' Double mutant tissue did not qualitatively differ in appearance under DIC microscopy prior to recordings, nor was the number of 'unhealthy' cells greater than that of other genotypes (7% vs. 4–11% of all cells across control genotypes at P400). Spatial comparison of neural activity was obtained by recording from serial sections in the flocculus, lateral (2nd or 3rd), intermediate (6th or 7th), and medial (11th or 12th) slices. Lower number slices were used in the younger age groups (P45 and P110) to roughly match the relative positioning of recordings across age groups. 0–3 recordings were made from each lobule within each slice dependent on tissue quality and health. Each recording lasted for 1 min. 3–5 mice were used for each age group, and the experimenter was blinded to the genotype, age, and sex.

Intracellular recordings were obtained from PNs in either lobule III or VIII of the medial cerebellum (i.e., vermis); no statistical differences in properties were observed between lobules.

### Analyses

Spontaneous AP interstimulus intervals were detected and analyzed using standard and custom routines in ClampFit (Molecular Device), IgorPro (Wavemetrics), and Excel (Microsoft). Specifically, APs were threshold detected and spiking statistics (i.e., frequency and interval length) were determined using adapted IgorPro routines (Taro Tools; https://sites.google.com/site/tarotoolsregister/). The coefficient of variation of the mean inter-spike interval (CV) and the median inter-spike interval (CV2=2 |ISIn+1−ISIn|/(ISIn +1 + ISIn)) were calculated in Excel using custom macros.

Standard membrane properties were analyzed using IgorPro. $R_M$ was determined by averaging three voltage trace responses to a –5 mV step pulse from a –80 mV holding potential and measuring the resulting current deflection between 900 and 1000 ms after onset. The membrane time constant was measured by fitting a single exponential to the initial decay phase from 90% to 10% of the peak. $C_M$ was calculated by dividing the membrane time constant by the $R_M$. sEPSC events were recorded over a 1-min epoch and detected and measured using Neuroexpress (v21.1.13). Parallel and climbing fiber axons were stimulated using theta-glass electrodes (W.P.I.) and a TTL-controlled stimulus isolator (ISO-Flex, A.M.P.I.). Evoked EPSC amplitudes and decay time constants (one exp. for parallel and two exp. for climbing fibers) were analyzed using custom routines in IgorPro. APs were examined as part of a set of 1 s current injections between –500 and 2250 pA (250 pA steps) with a holding current adjusted to maintain an ~70 mV potential. AP waveforms were measured using custom routines in IgorPro. AP threshold was defined as the first membrane voltage in which the first derivative exceeded 30 mV/ms (*Zhu et al., 2006*).

## Examination of cerebellar atrophy

### Cerebellar size
Immediately after brain removal from the skull, a dorsal, whole-mount image was obtained. Images were then processed using Fiji (NIH). The forebrain and cerebellar sizes were assessed by outlining their two-dimensional space and then calculating area. We normalized for possible differences in overall brain size by dividing the results of the cerebellum by forebrain size to produce a relative cerebellum-to-forebrain ratio. Experimenters were blind to the genotype of the animal.

### Immunohistochemistry
At the respective study endpoints (P45, P120, P210, and P400), male and female mice of all genotypes represented in this study were anesthetized with isoflurane and underwent transcardial perfusion with phosphate-buffered saline (PBS) followed by 4% (w/v) buffered paraformaldehyde (PFA) and then dissected to extract the brain. Images of the whole brain were taken immediately after removing the brain from the skull and the brains were then submerged in 4% PFA for 24 hr, and then cryoprotected in Tris-buffered saline (TBS) with 0.05% azide and 30% sucrose for 72 hr and stored at 4°C until further use. The cerebellum was separated from the forebrain and parasagittally sectioned using a sliding microtome (Microm HM 430, Thermo Fisher Scientific) set to section at 40 μm thickness. Cerebellum sections were collected in a series of six and stored in TBS-AF (TBS with 30% sucrose, 0.05% sodium azide, and 30% ethylene glycol) at 4°C or – 20°C until further use. For immunofluorescent visualization of PNs, cerebellum sections of both $Atm^{+/+}$; $Aptx^{+/+}$ and $Atm^{R35X/R35X}$; $Aptx^{-/-}$ (n=5 per genotype) were washed for 5 min in TBS three times, and then blocked in 15% normal goat serum at room temperature for 30 min followed by free-floating incubation in rabbit or mouse anti-calbindin D-28k (1:1000) for 1 hr at room temperature on an orbital shaker. Sections were then washed for 5 min with TBS three times, followed by free-floating incubation in goat anti-rabbit or mouse Alexa Fluor 488 (1:1000) for 1 hr in the dark at room temperature on an orbital shaker. Following secondary antibody incubation, sections were washed for 5 min in TBS three times, then mounted and cover-slipped with Fluoromount-G with DAPI. For some sections, anti-cleaved Caspase-3 (1:200) and anti-CD68 (1:400) antibodies were additionally probed in parallel with Calbindin using an Alexa Fluor 647 (1:500) secondary antibody. Slides were scanned using Stereo Investigator (MBF Bioscience, ver. 2020) on a Zeiss microscope equipped with an ApoTome 2 (Carl Zeiss Microscopy, Axio Imager.M2) using either a 2.5×, 10×, 20×, 40×, or 63× objective and images were captured with a Hamamatsu CMOS camera (Hamamatsu Photonics, ORCA Flash 4.0 LT+).

To quantify the number of calbindin-reactive cells in each lobule in the resulting images, we used Stereo Investigator to randomly draw two lines between 300 and 500 μm long in each lobule and manually counted the total number of PNs along the length within the 40 μm thickness of the tissue slice under 40× magnification. 2D density (# of PNs/(linear length*40 μm thickness)) of the two samples per lobule were then averaged for further comparison between lobules and animals.

Calbindin positive PN dendrite widths were measured at a predefined location in lobule VI from each animal in 25 or 40 μm thick tissue sections under 20× magnification. Dendritic widths of the primary and secondary branches were measured at the midline between the PN cell bodies and edge of the ML. Between 7 and 13 dendrites were measured per section, one section per animal.

For PN somatic measurements, Stereo Investigator was used to randomly select PNs distributed across the entire medial section under 20× magnification. The average PN width per animal was determined by averaging results across three serial sections (16–37 PNs per section). PN widths were measured perpendicular to the PN layer or to the exiting dendrite if askew by more than a few degrees.

ML and GCL (visualized with Calbindin and DAPI stains, respectively) widths were assessed in Stereo Investigator by averaging two width measurements at predefined locations for each lobule, roughly halfway along the long extent of each lobule under 2.5× magnification.

CD68 positivity in the cerebellar sections was quantified by measuring the total percent area of CD68⁺ positive staining across the entire medial cerebellar section. 10× stitched images were thresholded to the negative control and quantified using ImageJ, one section per animal.

To quantify the percent of Calbindin-positive PNs that were positive for cleaved Caspase-3 we counted PNs across the entire cerebellum using Stereo Investigator. Three, 20× magnification stitched images per animal were examined and the results averaged. The threshold for Caspase-3 positivity was established from control sections stained with only the secondary antibodies.

For non-fluorescent histological analysis, 25-μm-thick, free-floating tissue sections onto positively charged slides and air-dried overnight. The tissue was washed in PBS twice for 5 min, then stained sequentially with 0.1% Hematoxylin in 95% ethanol for ~25 s and 0.5% Eosin in 95% ethanol for ~3 s and washed in double-distilled water after each stain. The tissue was subsequently dehydrated for 1 min in 95% ethanol, 100% ethanol, and 100% Xylene washes, then cover slipped with Permount. Slides were imaged using a color camera (Q Imaging, MBF Biosciences) on the same Zeiss microscope and MBF acquisition software.

Experimenters were blinded to the mouse genotype in which sections were examined, and the order of examination was interleaved for all histological measurements.

## Flow cytometry measurements

Flow cytometry analysis of blood and thymus cells was performed by staining with specific anti-mouse antibodies: CD4, CD8, CD3, CD44, and CD25. Briefly, whole-blood samples (50 μl) were stained using fluorescent-labeled antibodies, then red blood cells were lysed using BD lysing solution while live white blood cells were stained using a viability stain. Thymi were mechanically dissociated. 1–2 million thymus cells were similarly stained using specific antibodies for CD4, CD8, CD44, and CD25. Analysis of immuno-stained white blood cells or thymus samples was performed using FACS ARIA III and data was analyzed using FlowJo software as reported previously (*Sanghez et al., 2017*).

## Western blots

Protein extracts (cells/tissues) were homogenized in radioimmunoprecipitation assay (RIPA) lysis buffer (150 mM NaCl, 1% Nonidet P-40 [NP-40], 0.5% deoxycholate, 0.1% SDS, 50 mM Tris, and pH 8.0) with protease inhibitors (10 μg/ml AEBSF, 10 μg/ml leupeptin, 5 μg/ml pepstatin, 5 μg/ml chymotrypsin, and 10 μg/ml aprotinin). The protein extracts were sonicated then pelleted by centrifugation at 13,000 rpm for 15 min at 4°C. BCA protein assay was used to quantify protein concentrations. Samples containing equal amounts of protein 50–100 μg per lane were separated using 4–12% gradient TGX precast gels Bio-Rad then transferred by TransBlot Semi-Dry Bio-Rad system using Nitrocellulose transfer pack. Transferred blots were stained by Ponceau S stain for equal protein loading then washed and blocked with 5% nonfat dry milk in TBST for 60 min at room temperature. Primary antibodies were incubated with shaking overnight at 4°C. Blots were probed for the following antibodies: ATM (D2E2) Rabbit mAb Cell Signaling, at 1:1000 dilution, β-Actin (D6A8) Rabbit mAb Cell Signaling, GAPDH (D16H11) Rabbit mAb Cell Signaling followed by the appropriate horseradish peroxidase-conjugated (HRP) secondary Anti-rabbit, Anti-mouse for 2 hr at room temperature. After multiple washes with TBST, protein expression was detected by Radiance Plus chemiluminescence substrate using the Azure c400 and the Bio-Rad ChemiDoc imaging systems. Densitometric analysis of the ATM was performed using ImageJ. Experiments were performed with 2 technical and 3–5 biological replicates as indicated.

## Statistical assessment

The number of animals chosen for each group was based on a priori power analyses using GPower v3.1 based on an α size of 0.5, power of 0.8, and effect sizes estimated from preliminary data or prior studies. We used both parametric (one- and two-way ANOVA) for normally distributed and nonparametric (Kruskal-Wallis) statistical methods for interval data to test for differences between groups followed by pairwise multiple comparisons tests as indicated in the text. Outliers for immune data in *Figures 6 and 7* were excluded via the ROUT method (Q=2%). The specific analyses used for each data set is noted in each figure legend. For all figures: ns=not significant, * p≤0.05, **p<0.01, ***p<0.001, ****p<0.0001. Data are reported as mean ± SEM and box and whisker plots indicate the minimum, first quartile, median, third quartile, and maximum data values. All figures and statistical analyses were completed using Excel (Microsoft 360) or Prism v8 and 9 (Graphpad).

## Acknowledgements

The authors would like to thank the UCLA Behavioral Core, especially Irina Zhuravka for her efforts assaying behavioral deficits in the mice. The authors would also like to thank Dr. Jennifer Fogel for her comments and edits to the manuscript.

## Additional information

### Funding

| Funder | Grant reference number | Author |
| --- | --- | --- |
| National Institute of Neurological Disorders and Stroke | R21NS108117 | Paul J Mathews |
| National Institute of Neurological Disorders and Stroke | R21NS108117-01S1 | Paul J Mathews |
| National Institute of Neurological Disorders and Stroke | R03NS103066 | Paul J Mathews |
| American Lebanese and Syrian Associated Charities of St. Jude Children's Hospital | | Peter McKinnon |
| National Institute of Neurological Disorders and Stroke | R01NS037956 | Peter McKinnon |
| National Cancer Institute | P01CA096832 | Peter McKinnon |
| National Center for Advancing Translational Sciences | UL1TR001881 | Paul J Mathews |
| Manitoba Mental Health Research Foundation | 312864 | Geoffrey G Hicks |
| Manitoba Health Research Council | 761023032 | Geoffrey G Hicks |
| Sparks | 13CAL01 | Richard A Gatti |
| National Institute of Neurological Disorders and Stroke | R33NS096044 | Michelina Iacovino |

The funders had no role in study design, data collection and interpretation, or the decision to submit the work for publication.

## Author contributions
Harvey Perez, May F Abdallah, Martin T Egeland, Investigation, Methodology, Supervision, Writing – review and editing; Jose I Chavira, Valentina Sanghez, Investigation, Methodology, Writing – review and editing; Angelina S Norris, Data curation, Formal analysis, Methodology, Writing – review and editing; Karen L Vo, Formal analysis, Investigation, Methodology, Writing – review and editing; Callan L Buechsenschuetz, Jeannie L Kim, Formal analysis, Writing – review and editing; Molly Pind, Developed and generated mice with 103C>T mutation, Methodology; Kotoka Nakamura, Methodology; Geoffrey G Hicks, Methodology, Resources, Writing – review and editing; Richard A Gatti, Conceptualization, Inventor of SMRT compounds, Methodology, Resources; Joaquin Madrenas, Conceptualization, Writing – review and editing; Michelina Iacovino, Conceptualization, Formal analysis, Visualization, Writing – review and editing; Peter J McKinnon, Conceptualization, Resources, Writing – review and editing; Paul J Mathews, Conceptualization, Formal analysis, Funding acquisition, Investigation, Methodology, Project administration, Resources, Supervision, Visualization, Writing – original draft, Writing – review and editing

## Author ORCIDs
Joaquin Madrenas (iD) http://orcid.org/0000-0001-6191-3733
Paul J Mathews (iD) http://orcid.org/0000-0002-1991-0798

## Ethics
Ethics StatementThis study was performed in strict accordance with the recommendations in the Guide for the Care and Use of Laboratory Animals of the National Institutes of Health. All the animals were handled according to approved Institutional Animal Care and Use Committee (IACUC) protocols at The Lundquist Institute (31374–03, 31773–02) and UCLA (ARC-2007–082, ARC-2013–068). The protocol was approved by the Committee on the Ethics of Animal Experiments of the Lundquist Institute (Assurance Number: D16-00213). Every effort was made to minimize pain and suffering by providing support when necessary and choosing ethical endpoints.

## Decision letter and Author response
Decision letter https://doi.org/10.7554/eLife.64695.sa1
Author response https://doi.org/10.7554/eLife.64695.sa2

---

# Additional files

## Supplementary files
• Transparent reporting form

## Data availability
All data generated or analyzed during this study are included in the manuscript and supporting source data files.

---

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
