## [Editor Report]

The authors have developed a new mouse model for Ataxia Telangiectasia. This mouse is of great value, which exhibits core features of the human disease including cerebellar Purkinje cells defects, cancer, and immune abnormalities. The potential value of the model is demonstrated using small molecular weight compounds to restore molecular function.

---

## [Decision Letter]

**Decision letter after peer review:**

Thank you for submitting your article "A novel, ataxic mouse model of Ataxia Telangiectasia caused by a clinically relevant nonsense mutation" for consideration by *eLife*. Your article has been reviewed by 3 peer reviewers, including Roy V Sillitoe as the Reviewing Editor and Reviewer #1, and the evaluation has been overseen by Huda Zoghbi as the Senior Editor. The following individual involved in review of your submission has agreed to reveal their identity: Alanna J Watt (Reviewer #3).

The reviewers have discussed the reviews with one another and the Reviewing Editor has drafted this decision to help you prepare a revised submission.

Summary:

In this manuscript, the authors have generated a new mouse model for the severe disease, Ataxia Telangiectasia (A-T). They introduce null mutations in Atm onto the background of mice that are somewhat sensitized since they also harbor mutations in the Aptx gene. The outcome is the mice show a set of phenotypes that are strikingly similar to symptoms seen in human patients. These include cerebellar degeneration, cancer, and immune system abnormalities. The also deliver small molecule readthrough (SMRT) compounds into tissue explants and show that such a manipulation can restore the production of ATM protein. The success in producing an Atm model with cerebellar degeneration is a compelling advance as this particular phenotype has been incredibly difficult to reproduce in animal models. The authors perform an interesting set of analyses to confirm that the other important features of the disease are also present in their mice. This paper has broad interest to multiple fields including neuroscience, cancer, and immunology.

Essential revisions:

1) It is not clear how progressive the cerebellar degeneration is. What is the spatiotemporal pattern of degeneration? Please consider the lobule by lobule effects over time.

2) For the electrophysiology, what stage cells have you recorded from? That is, what was the structure of the Purkinje cells that you recorded? If the cells look really "normal" but fire abnormally, then please comment on how they are being affected. If the morphology is abnormal, then please explain what defects you see and how they might impact function. Essentially, the authors need to disentangle cell autonomous effects and non-cell autonomous effects with more clarity. That is, are you studying the "dying" cells or the cells that escaped the genetic defect?

3) Are both the Atm and the Aptx genes expressed in all (or the same) Purkinje cells? What is the experimental evidence?

4) Please provide more context and rationale for Aptx in the abstract. As it stands, its mention comes out of nowhere.

5) In the Introduction, please provide more information as to why previous studies/models might have failed to produce severe Atm-related cerebellar phenotypes.

6) In the Introduction, the rationale for the choice of paring the Atm mutations with defects in the Aptx gene is unclear. Are they in the same pathway? Are the genes located in close proximity to one another? There are many issues that need to be discussed.

Related to above, ATM and APTX, while involved in DDR, are involved in parallel pathways-ATM in DNA double stranded break repair, and APTX in single stranded break repair. Homozygous mutations in APTX causes human ataxia (AOA1), but there is nothing to indicate an intersection mechanistically between AT and AOA1. One could just as well call the AT-APTX double mutation a model of AOA1. As indicated above, please expand on the rationale of the experimental design.

Also, are there more single stranded DNA breaks? Double stranded DNA breaks? Is there a sequestration of SS DNA break repair components including PARP1? How are the changes in PC firing related to DDR (it would be worthwhile for the authors to examine the following papers Hoch et al. Nature. 2017 Jan 5;541(7635):87-91, Stoyas et al. Neuron 2020 Feb 19;105(4):630-644) to give insight into studies that can explore mechanism for DDR and changes in cerebellar morphology/function.

Therefore, the authors need to address whether single vs double stranded break repair is present and the authors could do a better job of linking the change in PC firing to DNA damage.

7) Figure 2B: Apologies if I am missing something, but I do not understand the reason or explanation for what determines the probability of survival for the green, gold, and orange traces (the three severe cases in the graph). That is, why is the gold so strong?

8) How come rotarod was not used as a test? This is a standard motor behavior test that is useful for comparing across animal models and studies.

9) Related to above, why not use in vivo recordings? I can understand using slice recordings to tackle the biophysical and intrinsic mechanisms, although the authors did not do that. It seems to me that extracellular recordings would have been more informative in the in vivo, awake context.

10) The authors picked specific regions of the cerebellum to target their slice recordings, which is perfectly reasonable. But why did you pick these regions? Please provide a full justification and discussion for the importance of these particular lobules in relation to what you are trying to solve.

11) Given the use of slice recordings and that Purkinje cell degeneration is a key aspect of the phenotype, it would be very compelling if the authors showed some filled cells. As it stands, it is very hard to appreciate what the severity of neuropathology actually looks like, especially in relation to what the functional defects are teaching us.

12) The authors state that "The largest differences were detected in the anterior [38.6±3.4 Hz (n=187) vs. 88.1±1.8 Hz (n=222)] and posterior [46.9±1.9 Hz (n=175) vs. 84.1±2.4 Hz (n=219)] medial cerebellum [1-way ANOVA, p<0.0001; Figure 4B]." Okay, but what does this mean? What is your interpretation for why these regions were more heavily impacted (cell sensitivity based on circuit architecture, gene expression and protein make-up, neuronal lineage?) and how might it impact the phenotype?

13) The authors state and reference "Previous studies in mouse models of heritable ataxia indicate that physiological disruption in PN firing not only includes changes in frequency but also affects its regularity (Cook, Fields, and Watt 2020)." I agree with having this reference, but what about other models of ataxia? There are a number of other excellent models that should be discussed.

14) Purkinje cell firing data (figure 4B) should not be averaged across all of the ages, as this is not standard practice, and would be akin to averaging all behavior across ages. I think the data in Figure 4C suffices. If you want to compare across lobules on one graph, simply choose a particular age (perhaps when behavioral changes are first observed?) or at the oldest age.

15) Why examine Purkinje cell firing deficits in different lobules but not make that distinction for Purkinje cell loss? The Purkinje cell loss analysis focussed on the areas with most pronounced firing deficits but this means that we don't know whether the cells that fire abnormally are the only ones that die. Also see point #2 above.

16) Figure 4E and related text: Please provide a much more extensive set of images to show the cerebellar pathology. 1) Please show views of the different lobules to demonstrate the pattern of degeneration. 2) Please show different ages to show the progression of degeneration. 3) Please show higher power images of the Purkinje cells to clearly demonstrate their morphology.

17) The authors need to need provide more data for what is actually happening in relation to cell death. Why not perform Tunel or caspase staining etc.? The authors must show that there are actually acellular gaps where cells have died, or some other indication that cell death has occurred or is occurring.

18) Also in relation to the Purkinje cell degeneration, what do the dendrites look like? What about the axons? Do you see any torpedoes or axonal regression?

19) In regards to the cerebellar degeneration, what happens to the other cell types in the cerebellar cortex? Are they intact? What about the cerebellar nuclei?

20) The authors state "Of interest, APTX deficiency by itself had the greatest effect on the loss of DN4 cells…". Okay, but it is hard to see what this means for A-T as a disease. Interesting as it is, what is the relevance of this gene and these findings to the actual disease?

21) Please provide a more extensive description and rationale for why this explant system was chosen.

22) Figure 7: There have several comments regarding the blots. 1) The blots look like they have clipped too much. Please show the whole thing. 2) It looks as though the bands are slightly higher in the treated condition. Why? 3) What are the molecular weights? Please provide a ladder.

[Editors' note: further revisions were suggested prior to acceptance, as described below.]

Thank you for resubmitting your work entitled "A novel, ataxic mouse model of Ataxia Telangiectasia caused by a clinically relevant nonsense mutation" for further consideration by *eLife*. Your revised article has been evaluated by 2 peer reviewers, one of whom is a member of our Board of Reviewing Editors, and the evaluation has been overseen by Gary Westbrook as the Senior Editor.

The manuscript has been improved but there are some remaining issues that need to be addressed, as outlined below in the comments of Reviewers 1 and 3. The remaining concerns are related to the pathology of the mouse model. Specifically, the reviewers have noted several issues regarding the clarity of the presentation and quantification of the data demonstrating the impact of the genetic mutations on Purkinje cell structure.

*Reviewer #1:*

The authors have done an outstanding job of addressing most of the comments that were raised by adding new data and revising the text. Still, a number of issues remain.

The main concern with the current version is that despite an attempt to resolve the localization and extent of neuropathological defects in the mutant, the authors have now presented the data in a manner that raises concerns as to how severe the pathology actually is. Indeed the authors state that it is not severe, although without additional quantification it is challenging to fully understand what this means.

1) On page 13, it is unclear what the authors mean by "we do not qualitatively observe severe pathological changes in the anatomy of their cerebellum…" One way to deal with this issue is to provide quantifications for the data.

2) Related to above, the authors need to quantify whether Purkinje cell loss actually occurs. In its current form, the descriptions of the data are very confusing. If there is no cell death, then this must be described as a key aspect of the model. Similarly, the dendritic defects must be quantified and axonal swellings described and quantified in detail.

3) Is there a possibility that the axonal swelling that are labeled by the authors are just simply cut ends of Purkinje cell axon collaterals? How do you define these strictly as pathological swellings?

4) From the anatomical data provided, it is hard to appreciate the dendritic defects, especially at this low magnification.

5) Figure 5-supplement 5, panel C. The anatomical data needs equivalent images from controls. Also, the higher power images do not have a scale bar.

6) The authors state that "However, since the neuropathology observed in the AtmR35X/R35X; Aptx-/- mice is not severe, our findings suggest that like several SCAs (e.g.,1 and 3), changes in cerebellar function itself, rather than profound loss of cerebellar cells, is likely sufficient to cause the ataxic phenotype (Shakkottai et al. 2011b; Lorenzetti et al. 2000; Clark et al. 1997). I am confused by this statement because the neuropathology in SCA1 is in fact very severe.

*Reviewer #3:*

The authors should be commended for the revisions, which has substantially improved the manuscript. Overall, the paper has been strengthened, and the development of a mouse model for A-T seems to merit publication in *eLife*.

However, there are still a few small details that lack clarification.

1. The authors have removed their data on cell death, stating that by using the light microscope, they were able to determine that spaces lacking calbindin still contained cells, but that they are calbindin negative. This is surprising to me, because there is no obvious decrease in calbindin intensity evident in their images (e.g. Figure 5D, and Supplemental Figure 5.5). This result is important because counting calbindin-positive cells is a standard approach for assessing whether Purkinje cell death exists, and is often used in the study of mouse models of ataxia. Characterizing the Purkinje cell number – whether it is decreased or it is not – is important, and should be included in the manuscript. The authors state that they looked using light microscopic images, and are able to see cells that are calbindin-negative, but this is not a standard way of assessing whether Purkinje cells are intact or not. I would suggest that they count using another cellular marker, or a cell death marker. The authors need to show clearly how they are counting cells, and how they are sure that cells in the light microscope image reflect live, and not dead, cells.

2. The authors show evidence of swellings on Purkinje cell axons but do not show what the axons look like in their control mice in Supplemental Figure 5.5. Please add that.

3. The authors do not report any statistics for Supplemental Figure 5.1. Please add that.

4. While some improvement has been made in the justification for the data in Figure 8 to be included in the manuscript, it is still a poor fit. For example, these data are in a different mouse model that is not the focus of the rest of the paper, and has very low N's (n = 2 for one condition). The data does not flow well with the rest of the paper, and seems to really come out of the blue from the rest of the paper. The authors should either motivate this better, or, ideally, remove this data and include it in a subsequent paper.

5. Please edit lines 496-7. The statement that Cav2.1 is shown to contribute to baseline excitability is not shown in the cited paper.

---

## [Author Response]

Essential revisions:1) It is not clear how progressive the cerebellar degeneration is. What is the spatiotemporal pattern of degeneration? Please consider the lobule by lobule effects over time.

We now report measurements of the cerebellar molecular and granule cell layer over time and for each lobule. This is reported in Figure 5 and supplemental figures. Briefly, we find a progressive shrinking of the Purkinje neuron dendritic layer (i.e., molecular layer), but no change in the width of the granule cell layer. Atrophy appears to be a global phenomenon and not distinct to any specific lobule or medial to later subdivision. Given that anatomical subdivisions do not suitably correlate with differential molecular expression patterns of Purkinje neurons (e.g., aldolase C) and potentially not function either, we have made the decision to report, but not highlight or potentially over interpret these findings, as there is a good chance that anatomical subdivisions are an inadequate, although logistically straightforward way to group the data.

2) For the electrophysiology, what stage cells have you recorded from? That is, what was the structure of the Purkinje cells that you recorded? If the cells look really "normal" but fire abnormally, then please comment on how they are being affected. If the morphology is abnormal, then please explain what defects you see and how they might impact function. Essentially, the authors need to disentangle cell autonomous effects and non-cell autonomous effects with more clarity. That is, are you studying the "dying" cells or the cells that escaped the genetic defect?

As requested, we now provide a broader description of cell selection in both the results and methods section. However, since we didn’t find gross degeneration of the cerebellum our intra and extracellular recordings are likely to be representative of those contributing to the phenotype.

Briefly, only cells with a “healthy” look (low contrast of cellular borders), and regular, uninterrupted firing rate were examined. During analysis, a small fraction of cells that appeared to fire regularly prior to proceeding with recording were subsequently found to have large gaps (>2s) in their firing rate over the 60s recording. These cells were eliminated from analysis. Double mutant tissue did not qualitatively differ in appearance under DIC microscopy prior to recordings, nor was the number of “non regular” firing cells in the double mutant mice greater than that of other genotypes (7% vs 4-11% of all cells across control genotypes at P400).

3) Are both the Atm and the Aptx genes expressed in all (or the same) Purkinje cells? What is the experimental evidence?

This is an excellent question, as several proteins have been demonstrated to be expressed in Purkinje neurons in parasagittal bands across the cerebellum. It is generally thought that ATM and APTX genes are ubiquitous throughout the nervous system and peripheral tissues. Informal discussions with other experts on these proteins indicate that they have not observed differential expression patterns across the cerebellum. However, selective antibodies for ATM and APTX have historically been a problem in the field, and the relatively low abundance of ATM and APTX makes them a challenge for histological examination. Unfortunately, we tested several antibodies for immunofluorescence but were unable to validate any suitable for a quantitative or even qualitative analysis of expression levels across the cerebellum.

4) Please provide more context and rationale for Aptx in the abstract. As it stands, its mention comes out of nowhere.

We have rewritten the abstract so APTX is less of a surprise.

5) In the Introduction, please provide more information as to why previous studies/models might have failed to produce severe Atm-related cerebellar phenotypes.

We have significantly expanded the introduction to better detail potential reasons for why prior models failed to produce ataxia.

“Unfortunately, none develop an overt, progressive ataxia with cerebellar dysfunction and atrophy that recapitulates the human disease. […] Other challenges include potentially leaky genetic manipulations that result in low levels of ATM protein or active fragments with residual kinase activity, thus limiting neuropathology (Li et al. 2011).”

6) In the Introduction, the rationale for the choice of paring the Atm mutations with defects in the Aptx gene is unclear. Are they in the same pathway? Are the genes located in close proximity to one another? There are many issues that need to be discussed.

To make this clearer we now provide a more thorough rationale detailing why we chose to pair APTX loss, with that of ATM. The following paragraph has been added to the introduction:

“We test here whether increasing genotoxic stress, by placing null mutations in not just the Atm gene, but also the related Aptx gene, leads to a more representative mouse model that displays cerebellar dysfunction, atrophy, and the development of progressive ataxia. […] We therefore reasoned that deficiency of both proteins would have an additive effect on genotoxic stress capable of inducing a detectable neurological dysfunction.”

Related to above, ATM and APTX, while involved in DDR, are involved in parallel pathways-ATM in DNA double stranded break repair, and APTX in single stranded break repair. Homozygous mutations in APTX causes human ataxia (AOA1), but there is nothing to indicate an intersection mechanistically between AT and AOA1. One could just as well call the AT-APTX double mutation a model of AOA1. As indicated above, please expand on the rationale of the experimental design.Also, are there more single stranded DNA breaks? Double stranded DNA breaks? Is there a sequestration of SS DNA break repair components including PARP1? How are the changes in PC firing related to DDR (it would be worthwhile for the authors to examine the following papers Hoch et al. Nature. 2017 Jan 5;541(7635):87-91, Stoyas et al. Neuron 2020 Feb 19;105(4):630-644) to give insight into studies that can explore mechanism for DDR and changes in cerebellar morphology/function.Therefore, the authors need to address whether single vs double stranded break repair is present and the authors could do a better job of linking the change in PC firing to DNA damage.

While APTX was initially linked to single stranded DNA repair, recent evidence suggests a role in both single and double stranded repair. This is now better described in the introduction. Given that there isn’t a clear-cut separation of ATM and APTX in double and single stranded DNA repair we do not believe that direct comparison of these processes in the cerebellum would be as illuminating as if this were the case.

While we generally agree that from a neurological standpoint the mouse we have generated might interchangeably be described as an A-T or AOA1 mouse model, the other symptomology displayed by the mouse, including the increased cancer prevalence and perturbations to the immune system are characteristic of A-T and not AOA1. Therefore, in total, we believe this mouse to be a more representative model of A-T and not AOA1.

The goal of linking PC or even cerebellar dysfunction to DNA damage is of great importance, and of extremely high impact. We believe that this model will provide a tool to address this question, but it will require significant directed efforts outside our initial, comprehensive description of the new mouse model in this manuscript.

7) Figure 2B: Apologies if I am missing something, but I do not understand the reason or explanation for what determines the probability of survival for the green, gold, and orange traces (the three severe cases in the graph). That is, why is the gold so strong?

As requested, we now provide additional details to significantly improve clarity. The simple answer here is that ATM deficiency, regardless of the amount of APTX present, results in premature death. While qualitatively the R35X/R35X;+/- (gold) animals appear to have a more severe phenotype than the R35X/R35X;+/+ (green) there is no significant difference between the three genotypes fully deficient in ATM (gold, orange, and green). This last point has now been explicitly stated in the figure legend to clarify this point. Due to the number of groups plotted, error bars were deliberately left off as they made it extremely difficult to make out each group. The following text was added to Figure 3B:

**“**(B) ATM deficient mice, regardless of APTX expression displayed significantly lower survivability with ~55% of mice deceased by P400. […] Total number of animals indicated in panel C.”

8) How come rotarod was not used as a test? This is a standard motor behavior test that is useful for comparing across animal models and studies.

We chose to use the Catwalk because it provides a more quantitative measure of animal locomotion.

9) Related to above, why not use in vivo recordings? I can understand using slice recordings to tackle the biophysical and intrinsic mechanisms, although the authors did not do that. It seems to me that extracellular recordings would have been more informative in the in vivo, awake context.

Our goal to characterize several control genotypes, anatomical regions, and time points made in vivo, awake recordings impractical. We agree that future in vivo studies will be informative now that we have confirmed that at least one copy of the wildtype ATM or APTX gene is sufficient to rescue the phenotype! Moreover, we have now added an extensive biophysical examination of cerebellar neurons at P400, shedding light on the mechanisms of cerebellar dysfunction caused by ATM and APTX deficiency. We focus on Purkinje neurons here but include experiments that suggest inputs from both parallel fibers and climbing fibers remain relatively normal, suggesting Purkinje neurons as a potential focal point of the disease within the cerebellum. We have added an entire figure (Figure 4) and section that defines how loss of ATM and APTX changes the intrinsic properties of Purkinje neurons.

10) The authors picked specific regions of the cerebellum to target their slice recordings, which is perfectly reasonable. But why did you pick these regions? Please provide a full justification and discussion for the importance of these particular lobules in relation to what you are trying to solve.

We fully appreciate the reviewers desire for a more comprehensive justification and interpretation of the differential findings of Purkinje neuron action potential rates we report across anatomical regions of the cerebellum. However, as discussed in critique 1, we have taken a deliberately cautious approach in interpreting these data due to the fields incomplete understanding of cerebellar functional anatomy and methods for discriminating between domains. However, we have taken efforts to better rationalize why we examined the regions we did in the Results section and point out some of the limitations of using anatomy as a grouping method in the Discussion section.

11) Given the use of slice recordings and that Purkinje cell degeneration is a key aspect of the phenotype, it would be very compelling if the authors showed some filled cells. As it stands, it is very hard to appreciate what the severity of neuropathology actually looks like, especially in relation to what the functional defects are teaching us.

We now provide several high power (63X) images in Figure 5—figure supplement 5 of the Purkinje neurons. While these images do not contain single cell fills, they do fully illustrate the morphology of Purkinje neurons in the double mutant mice. These images highlight a common finding of thicker Purkinje neuron dendrites in the double mutant mice and some evidence of axonal swellings.

12) The authors state that "The largest differences were detected in the anterior [38.6±3.4 Hz (n=187) vs. 88.1±1.8 Hz (n=222)] and posterior [46.9±1.9 Hz (n=175) vs. 84.1±2.4 Hz (n=219)] medial cerebellum [1-way ANOVA, p<0.0001; Figure 4B]." Okay, but what does this mean? What is your interpretation for why these regions were more heavily impacted (cell sensitivity based on circuit architecture, gene expression and protein make-up, neuronal lineage?) and how might it impact the phenotype?

The reviewer raises several important and interesting questions that we look to explore in future studies as the experiments to appropriately address them are out of the scope of this study. As indicated in comment 10, we have expanded our discussion on these findings and the limitations of using anatomical groupings.

13) The authors state and reference "Previous studies in mouse models of heritable ataxia indicate that physiological disruption in PN firing not only includes changes in frequency but also affects its regularity (Cook, Fields, and Watt 2020)." I agree with having this reference, but what about other models of ataxia? There are a number of other excellent models that should be discussed.

The review article we cite, published in 2020, provides a comprehensive description of prior research that directly relates intrinsic PN firing differences across a broad array of over 15 models of ataxia, this includes several SCA variants, EA2, ASACS, as well as other disorders like Huntington’s and ASD. We now directly cite those research articles at the location of the noted quote. We have also improved the language within the sentence to make it clearer that this effect on regularity has been observed across several different ataxia models. However, we didn’t find that the regularity of firing in the “healthy” cells of our new A-T model to be significantly impacted and therefore do not address this phenomenon further in the article.

14) Purkinje cell firing data (figure 4B) should not be averaged across all of the ages, as this is not standard practice, and would be akin to averaging all behavior across ages. I think the data in Figure 4C suffices. If you want to compare across lobules on one graph, simply choose a particular age (perhaps when behavioral changes are first observed?) or at the oldest age.

As noted by the reviewer, we have removed this panel entirely.

15) Why examine Purkinje cell firing deficits in different lobules but not make that distinction for Purkinje cell loss? The Purkinje cell loss analysis focussed on the areas with most pronounced firing deficits but this means that we don't know whether the cells that fire abnormally are the only ones that die. Also see point #2 above.

Please see responses to 1, 2, and 10.

16) Figure 4E and related text: Please provide a much more extensive set of images to show the cerebellar pathology. 1) Please show views of the different lobules to demonstrate the pattern of degeneration. 2) Please show different ages to show the progression of degeneration. 3) Please show higher power images of the Purkinje cells to clearly demonstrate their morphology.

We have added several new images to Figure 5 and Figure 5—figure supplement 5. We now report a developmental shrinking of the width of the molecular layer, but not the granule cell layer (Figure 5D).

17) The authors need to need provide more data for what is actually happening in relation to cell death. Why not perform Tunel or caspase staining etc.? The authors must show that there are actually acellular gaps where cells have died, or some other indication that cell death has occurred or is occurring.

The reviewers desire to confirm cell death is an important one. After further inspection of the tissue, including inspection under brightfield rather than just fluorescent microscopy, it does not appear that the acellular gaps we initially observed under fluorescence resulted from cellular loss, but insufficient Calbindin staining. We performed a more intensive 3 dimension counting of Purkinje neurons across all medial folia and did not find a significant loss of PNs in the double mutant mice. Thus, we do not report or believe there is significant cell loss or death.

18) Also in relation to the Purkinje cell degeneration, what do the dendrites look like? What about the axons? Do you see any torpedoes or axonal regression?

We find the primary dendrites in the double mutant mice to qualitatively have a larger gauge than those found in wildtype animals. Although not found extensively throughout the tissue, we do find axonal swellings in the double mutant mice. These findings are illustrated in Figure 5—figure supplement 5.

19) In regards to the cerebellar degeneration, what happens to the other cell types in the cerebellar cortex? Are they intact? What about the cerebellar nuclei?

Overall, we find anatomical effects restricted to a progressive reduction in the width of the molecular layer (see Figure 5). We do not find a similar shrinking of the granule cell layer. These results are in line with our finding that synaptic inputs from the granule cells onto Purkinje neurons appear normal based on the electrophysiological data presented in Figure 4. The cerebellar nuclei were not examined extensively, but we observed no obvious gross changes in anatomy.

20) The authors state "Of interest, APTX deficiency by itself had the greatest effect on the loss of DN4 cells…". Okay, but it is hard to see what this means for A-T as a disease. Interesting as it is, what is the relevance of this gene and these findings to the actual disease?

The reviewers are correct, this result does not directly relate to A-T. However, our intent in pointing out this finding is twofold. First, we want to alert the more immunologically astute reader to the fact that this effect did not go unnoticed by our group, a finding that, to our knowledge, illustrates for the first time a selective defect in double negative thymocyte development. Second, we are “planting a flag” on this original finding to facilitate our current efforts defining how APTX involved in gene rearrangement during thymocyte development.

21) Please provide a more extensive description and rationale for why this explant system was chosen.

As requested, we now provide a more comprehensive discussion on why we chose to use the explant system. In short, our goal was to demonstrate, in proof-of-principle experiments that the nonsense mutation inserted into these mice can be overcome using pharmacological readthrough agents. We used explants, rather than in vivo delivery to mitigate any challenges that can arise with drug delivery (e.g., inability to cross the blood brain barrier). We have modified section 2.7:

“Our primary rationale for inserting a clinically relevant nonsense mutation in the Atm gene was to generate a mouse amenable to critical pre-clinical testing of a novel set of small molecule readthrough (SMRT) compounds. […] ATM expression was measured in samples from the spleen, where ATM is normally expressed at high levels, and the cerebellum, a key target tissue for the disorder.”

22) Figure 7: There have several comments regarding the blots. 1) The blots look like they have clipped too much. Please show the whole thing. 2) It looks as though the bands are slightly higher in the treated condition. Why? 3) What are the molecular weights? Please provide a ladder.

We do not find evidence that the treated protein is higher on the gel. We now provide a ladder and full ponceaus image to provide further information. ATMs molecular weight of 350 kDa is now shown to the right of the images.

[Editors' note: further revisions were suggested prior to acceptance, as described below.]

The manuscript has been improved but there are some remaining issues that need to be addressed, as outlined below in the comments of Reviewers 1 and 3. The remaining concerns are related to the pathology of the mouse model. Specifically, the reviewers have noted several issues regarding the clarity of the presentation and quantification of the data demonstrating the impact of the genetic mutations on Purkinje cell structure.

We thank the reviewers for the opportunity to further clarify the results of our manuscript. In this revision we provide additional quantification and new data on the histopathology of the A-T mouse model that bolsters our findings as requested by the reviewers. To strengthen our prior results indicating that cell death is not a feature of the mouse model, we roughly doubled the number of cerebella in which we quantified Purkinje neuron density. In total, these data indicate that there is no significant difference in the density of calbindin positive Purkinje neurons in either the whole vermis (wildtype: 4.5 ± 0.2 (n=9) vs. A-T mouse: 4.5 ± 0.3 (n=7), Welch’s t-test, p=0.9) or in any single lobule of the A-T mouse model (see Figure 5—figure supplement 5; p>0.6 for each lobule II through X). Supporting this finding are additional data that demonstrate the lack of increased expression of either Caspase-3 (cell death marker) or CD68 (microglial activation marker) as compared to wildtype and a positive control (mouse model of MPS III). These data are provided in Figure 5—figure supplement 2. As requested by reviewer 1, the lack of PN cell death is now better highlighted. We have added a panel to Figure 5 that includes cerebellar images at both 2.5x and 20x as well as a summary plot of PN density. We have also added images of cerebellar sections stained with Hematoxylin and Eosin (H & E) to provide an additional view of cerebellar histology, as suggested by reviewer #3. We also now report in Figure 5 data demonstrating changes in PN morphology in the A-T mouse as compared to control (i.e., smaller PN somatic width and increased dendritic caliber). Additional modifications to the manuscript have been made to address each of the reviewer’s specific comments.

Reviewer #1:The authors have done an outstanding job of addressing most of the comments that were raised by adding new data and revising the text. Still, a number of issues remain.The main concern with the current version is that despite an attempt to resolve the localization and extent of neuropathological defects in the mutant, the authors have now presented the data in a manner that raises concerns as to how severe the pathology actually is. Indeed the authors state that it is not severe, although without additional quantification it is challenging to fully understand what this means.1) On page 13, it is unclear what the authors mean by "we do not qualitatively observe severe pathological changes in the anatomy of their cerebellum…" One way to deal with this issue is to provide quantifications for the data.2) Related to above, the authors need to quantify whether Purkinje cell loss actually occurs. In its current form, the descriptions of the data are very confusing. If there is no cell death, then this must be described as a key aspect of the model. Similarly, the dendritic defects must be quantified and axonal swellings described and quantified in detail.

As requested, we have now quantified these data, demonstrating that Purkinje neuron cell death is not a key aspect of this model, but morphological changes are (see opening paragraph and Figure 5 for details).

3) Is there a possibility that the axonal swelling that are labeled by the authors are just simply cut ends of Purkinje cell axon collaterals? How do you define these strictly as pathological swellings?

The reviewers are correct and reference to these swellings has been removed.

4) From the anatomical data provided, it is hard to appreciate the dendritic defects, especially at this low magnification.

Quantification of the large-caliber dendrites has been added. Images have been added to Figure 5E, including insets of the dendrites at an increased magnification.

5) Figure 5-supplement 5, panel C. The anatomical data needs equivalent images from controls. Also, the higher power images do not have a scale bar.

We believe that our labeling of the controls in this panel was potentially unclear. This result has now been quantified and moved to the main Figure 5. All images contain the appropriate scale bars.

6) The authors state that "However, since the neuropathology observed in the AtmR35X/R35X; Aptx-/- mice is not severe, our findings suggest that like several SCAs (e.g.,1 and 3), changes in cerebellar function itself, rather than profound loss of cerebellar cells, is likely sufficient to cause the ataxic phenotype (Shakkottai et al. 2011b; Lorenzetti et al. 2000; Clark et al. 1997). I am confused by this statement because the neuropathology in SCA1 is in fact very severe.

We thank the reviewer for pointing out how this sentence might be confusing. Our intent was to support the idea that ataxia can arise from changes in PN physiology alone, either before, or even without cell death. While there is PN loss in both SCA1 and 3, as pointed out by the reviewer, behavioral defects are observed in both prior to detected neuronal loss. This is important, as it suggests that there may be an opportunity for therapeutic intervention in disorders of ataxia even after behavioral defects are detected. To strengthen this point further, we also add the reference Jayabal et al. 2016, which demonstrates behavioral defects in a mouse model of SCA6 prior to PN cell loss. The sentence has been changed to:

“Our histological observations in the Atm^R35X/R35X^; Aptx^-/-^ mice suggest that changes in cerebellar function itself, rather than profound loss of cerebellar cells, are sufficient to cause the ataxic phenotype, consistent with the observation of behavioral defects prior to significant PN loss in several SCAs (Shakkottai et al. 2011b; Lorenzetti et al. 2000; Clark et al. 1997).”

Reviewer #3:The authors should be commended for the revisions, which has substantially improved the manuscript. Overall, the paper has been strengthened, and the development of a mouse model for A-T seems to merit publication in eLife.However, there are still a few small details that lack clarification.1. The authors have removed their data on cell death, stating that by using the light microscope, they were able to determine that spaces lacking calbindin still contained cells, but that they are calbindin negative. This is surprising to me, because there is no obvious decrease in calbindin intensity evident in their images (e.g. Figure 5D, and Supplemental Figure 5.5). This result is important because counting calbindin-positive cells is a standard approach for assessing whether Purkinje cell death exists, and is often used in the study of mouse models of ataxia. Characterizing the Purkinje cell number – whether it is decreased or it is not – is important, and should be included in the manuscript. The authors state that they looked using light microscopic images, and are able to see cells that are calbindin-negative, but this is not a standard way of assessing whether Purkinje cells are intact or not. I would suggest that they count using another cellular marker, or a cell death marker. The authors need to show clearly how they are counting cells, and how they are sure that cells in the light microscope image reflect live, and not dead, cells.

We understand the reviewer’s surprise. In fact, we initially hypothesized that ataxia would be associated with a significant loss of PNs. However, we did not observe the significant acellular gaps or thinning of PNs in the cerebellum as observed with other mouse models of ataxia. To affirm our initial findings, we doubled the number of animals examined. Moreover, PN density was established using the standard method of dividing the number of calbindin positive PNs by the area examined. We did not find a significant difference in density either within the medial section as a whole or when we examined each lobule individually. We would also like to note that a report currently online as a preprint that examined PN density in a small set of A-T patients (n=9) found significant reductions in PN density only in late-stage, older patients (>25 years of age), with little to no change in patients between 16-25 years of age medRxiv: doi.org/10.1101/2021.01.22.20245217. The lack of quantitative studies, especially in younger patients, which is limited due to the rarity of the disorder and availability of postmortem tissues, has potentially resulted in an inaccurate impression based on the late-stage clinical reports that PN cell death is a driving factor of ataxia in A-T. We acknowledge that had we been able to look at mice out as far as 2 years of age, we may have found prominent cell death; however, it appears clear, that ataxia in this mouse model is not related to PN cell death. We further support this finding with additional data that demonstrates a lack of significant Caspase-3 or CD68 expression in the P400 Atm^R35X/R35X^; Aptx^-/-^ mice.

2. The authors show evidence of swellings on Purkinje cell axons but do not show what the axons look like in their control mice in Supplemental Figure 5.5. Please add that.

See response to Reviewer 1, Comment 3.

3. The authors do not report any statistics for Supplemental Figure 5.1. Please add that.

Demarcation of statistical significance has been added to the figure.

4. While some improvement has been made in the justification for the data in Figure 8 to be included in the manuscript, it is still a poor fit. For example, these data are in a different mouse model that is not the focus of the rest of the paper, and has very low N's (n = 2 for one condition). The data does not flow well with the rest of the paper, and seems to really come out of the blue from the rest of the paper. The authors should either motivate this better, or, ideally, remove this data and include it in a subsequent paper.

These mice were specifically designed to test our small molecule readthrough compounds. We have modified the introduction to make this rationale clearer.

“Finally, we designed this new mouse model to test our recently developed Small Molecule Read-Through Compounds (SMRT) that enable translation through premature termination codons (Du et al. 2013). Thus, we inserted a premature termination-causing nonsense mutation (103C>T) in the *Atm* gene common to a large family of North African A-T patients (Gilad, Bar-Shira, et al. 1996). This mutation results in a premature termination codon (PTC) at what would normally be amino acid 35 and the loss of ATM translation. Here, we report proof-of-principle experiments demonstrating that genetic mutations incorporated into the A-T mouse model are amenable to read-through compounds and thus appropriate for preclinical testing of the SMRT compounds.”

Additionally, we have added another biological replicate to the experiment in Figure 8; thus, all experiments are confirmed in at least triplicate.

5. Please edit lines 496-7. The statement that Cav2.1 is shown to contribute to baseline excitability is not shown in the cited paper.

Upon further reflection, we believe that this sentence suggests that for a majority of SCAs the link between the disease-causing protein defect and ataxia is straightforward. Since this is not the case, we have simplified this sentence to:

“The mechanistic link between deficiency in DNA stability proteins like ATM and APTX and PN dysfunction is far from clear.”